# Rnd3 regulates lung cancer cell invasion and migration independently of ROCK1 signaling via alpha 5 integrin modulation

Noemi Garcia Garcia[1,*], Thanh Ha Vy Nguyen[1,*], Dane Richey[1], Emily F Cox[1], Jon Coca Juaristi[1], Cody Ashby[2], Analiz Rodriguez[3], Katie R Ryan[1]

**Rnd3 regulates cellular processes commonly dysregulated in cancer, and altered Rnd3 expression has been linked to several cancer types. Here, we show lung adenocarcinoma patients expressing low levels of Rnd3 have significantly higher survival rates. To gain mechanistic insight into this correlation, we knocked down Rnd3 in lung adenocarcinoma A549 and H460 cell lines and patient-derived lung-to-brain metastasis (PDLBM) cell lines as a proxy for advanced disease. Depletion of Rnd3 expression decreases cell invasion and migration, two hallmarks of metastasis, independently of RhoA-ROCK1 signaling in both lung adenocarcinoma and PDLBM cell lines, indicating the involvement of a novel pathway. The expression of alpha 5 integrin was increased in Rnd3-depleted A549 cells, and knocking down alpha 5 integrin restored cell migration and invasion rates in A549 cells. Identification of a RhoA-ROCK1-independent mechanism by which Rnd3 modulates alpha 5 integrin expression to control cell migration and invasion provides new insights into the molecular basis of pro-malignant properties. These properties that drive the metastatic potential of lung adenocarcinoma may be exploited to target metastatic disease.**

## Introduction

Lung cancer is the leading cause of cancer deaths for both males and females in the United States (Siegel et al, 2021). Worldwide, there are an estimated 1.8 million deaths attributed to lung cancer each year (Bray et al, 2018), and the 5-yr relative survival rate is ~21% (Siegel et al, 2021). Lung cancer is generally subdivided into two histological subtypes, small-cell lung cancer, occurring at ~15%, and non–small-cell lung cancer, occurring at ~85% (Herbst et al, 2018). The most common subtype of non–small-cell lung cancer is adenocarcinoma, which comprises ~40–50% of all lung cancers (Travis et al, 2011; Duma et al, 2019). Most lung cancers metastasize, resulting in more complex disease that is harder to treat, and have poor prognosis (Siegel et al, 2021). Patients with lung cancer have an ~50% lifetime risk of developing brain metastasis, which causes the highest rate of mortality in lung cancer patients (Chi & Komaki, 2010; Yousefi et al, 2017). With the high incidence and poor prognosis of lung cancer, more research is required to identify key regulators of metastasis and new therapeutic targets.

Rho-family GTPases act as molecular switches and are master regulators of many aspects of cellular behavior (Etienne-Manneville & Hall, 2002; Jaffe & Hall, 2005; Hodge & Ridley, 2016). Cellular processes regulated by Rho GTPases are commonly dysregulated in cancer, with many studies focusing on what roles these proteins play in tumorigenesis. Like other members of the Rho GTPase family, Rnd3, also known as RhoE, has been implicated in the regulation of proliferation, cell adhesion, apoptosis, and cell migration (Guasch et al, 1998; Riento et al, 2003; Villalonga et al, 2004; Bektic et al, 2005; Liebig et al, 2009; Ryan et al, 2012). Rnd3 is an atypical Rho GTPase because of its inability to hydrolyze GTP; consequently, Rnd3 is not regulated by the Rho GTPase cycle but by alternative mechanisms (e.g., transcriptional regulation, posttranslational modification, and subcellular localization) (Foster et al, 1996; Riento et al, 2005b). Expression levels of Rnd3, rather than Rnd3 mutations, have been linked to cancer. High expression of Rnd3 increases cellular migration and invasion in several cancers (Klein & Aplin, 2009; Zhou et al, 2011, 2013; Feng et al, 2013), whilst depletion of Rnd3 decreases proliferation and cell migration (Klein et al, 2008; Klein & Aplin, 2009; Katiyar & Aplin, 2011; Feng et al, 2013; Clarke et al, 2015). In addition, Rnd3 has been shown to be overexpressed in lung cancer compared with normal tissue (Cuiyan et al, 2007; Zhang et al, 2007; Tsay et al, 2015; Li et al, 2016; Tang et al, 2018). Our analysis of lung adenocarcinoma patients, along with other groups' analyses of distinct lung cancer patient cohorts, shows the overall survival for lung cancer patients

---

[1]Department of Biochemistry and Molecular Biology, University of Arkansas for Medical Sciences, Little Rock, AR, USA   [2]Department of Biomedical Informatics, University of Arkansas for Medical Sciences, Little Rock, AR, USA   [3]Department of Neurosurgery, University of Arkansas for Medical Sciences, Little Rock, AR, USA

Correspondence: krryan@uams.edu
*Noemi Garcia Garcia and Thanh Ha Vy Nguyen contributed equally to this work

with high Rnd3 expression was significantly lower than that for patients with low Rnd3 expression (Zhang et al, 2007; Raz et al, 2008; Li et al, 2016; Sun et al, 2016).

Cell migration and invasion are regulated by a plethora of signaling pathways controlling different processes such as reorganization of the actin cytoskeleton and cell adhesion to the extracellular matrix (Trepat et al, 2012). Rnd3 regulates the actin cytoskeleton in many cell types, with Rnd3 overexpression most commonly associated with the loss of actin stress fibers and increased migration (Guasch et al, 1998; Hansen et al, 2000; Riento et al, 2003, 2005b). Previous studies have shown Rnd3 expression alters the actin cytoskeleton and cell migration via the RhoA-ROCK1 signaling pathway (Guasch et al, 1998; Riento et al, 2003, 2005a; Wennerberg et al, 2003; Klein et al, 2008; Pinner & Sahai, 2008; Klein & Aplin, 2009; Klein & Higgins, 2011). Rnd3 inhibits RhoA and its downstream effector, the serine/threonine kinase ROCK1, via two mechanisms, by recruiting p190RhoGAP to RhoA and by binding directly to ROCK1, both inhibiting downstream signaling (Guasch et al, 1998; Wennerberg et al, 2003). ROCK1, in turn, can phosphorylate Rnd3, increasing Rnd3 levels, resulting in a negative feedback loop (Komander et al, 2008; Riou et al, 2013). Depletion of Rnd3 leads to ROCK1 activation and the formation of actin stress fibers and focal adhesions, anchoring cells to the extracellular matrix and decreasing migration and invasive outgrowth (Klein et al, 2008; Klein & Aplin, 2009). Rnd3 has also been shown to regulate cell adhesion to the extracellular matrix by regulating beta 1 integrin activation, which leads to increased cell adhesion (Liebig et al, 2009; Endzhievskaya et al, 2023) and decreased directional cell migration (Endzhievskaya et al, 2023).

In this study, we identified a correlation between low Rnd3 expression and increased survival probability of lung adenocarcinoma patients. Rnd3 has been proposed to regulate many different cellular processes that are commonly dysregulated in cancer (Guasch et al, 1998; Riento et al, 2003; Villalonga et al, 2004; Bektic et al, 2005; Liebig et al, 2009; Ryan et al, 2012). To identify the mechanistic basis by which Rnd3 expression levels could affect patient survival, we identified the biological processes that were affected by knocking down Rnd3 expression in lung adenocarcinoma cell lines. We observed a significant decrease in cell invasion and cell migration, two hallmarks of metastasis, but no effect on cell proliferation or cell death. These data suggest that Rnd3 may regulate the metastatic potential of lung adenocarcinoma cells. With lung-to-brain metastasis being the leading cause of mortality of lung cancer patients (Chi & Komaki, 2010; Yousefi et al, 2017), we investigated whether Rnd3 signaling still functioned in patients with advanced metastatic disease. We used patient-derived lung-to-brain metastasis (PDLBM) cells as a proxy for advanced disseminated disease and observed that knocking down Rnd3 expression in these cell lines resulted in a significant decrease in cell invasion, indicating this pathway is still responsive. We next investigated the signaling pathways involved in Rnd3's regulation of cell invasion and migration. Surprisingly, we show that Rnd3's regulation of both cell invasion and migration is independent of ROCK1, a known downstream target of Rnd3 and modulator of Rnd3-regulated cell migration (Riento et al, 2003, 2005a; Klein et al, 2008; Klein & Aplin, 2009), as neither chemical inhibition nor knockdown of ROCK1 expression rescued the Rnd3-

knockdown phenotype, indicating a novel signaling pathway for Rnd3. In addition, we show that loss of RhoA expression was also unable to rescue cell migration or invasion back to wild-type (WT) rates in Rnd3-depleted lung adenocarcinoma cells. We investigated whether cell adhesion pathways were involved in Rnd3's regulation of cell migration and invasion, as Rnd3 has been shown to regulate integrin activation (Liebig et al, 2009; Endzhievskaya et al, 2023). We observed an increase in alpha 5 integrin expression in Rnd3-knocked down A549 cells, and knocking down alpha 5 integrin expression in these cells restored migration and invasion rates back to WT. We conclude that high Rnd3 expression correlates with poor prognosis in lung adenocarcinoma patients and that Rnd3 plays a critical role in cell invasion and migration (two key hallmarks of metastatic behavior) in both lung adenocarcinoma and PDLBM cells, indicating Rnd3 maintains control over these processes even in advanced disease. Furthermore, this occurs via a RhoA-ROCK1-independent pathway in lung adenocarcinoma and PDLBM cell lines and instead occurs through modulating integrin alpha 5 expression.

# Results

## Low expression levels of Rnd3 increase survival probability in lung adenocarcinoma patients

We analyzed transcript data and survival rates specifically for lung adenocarcinoma patients (N = 501) from TCGA-LUAD (The Cancer Genome Atlas Lung Adenocarcinoma). We grouped the patients based on Rnd3 expression, high or low. Using the optimal cut point, we identified ~34% (N = 172/501) with high Rnd3 expression and 66% (N = 329/501) with low Rnd3 expression (Figs 1 and S1B). We also used cutoffs based on the top (high) and bottom (low) quartile of high (N = 126/501) and low (N = 126/501) Rnd3 expression (Fig S1A and B). We then generated Kaplan–Meier plots for these groups and found that patients with low levels of Rnd3 have significantly higher survival probability (optimal cutoff $P$ = 0.00073, quartile cutoff $P$ = 0.0048) than patients with high levels of Rnd3 (Figs 1 and S1A). Our analysis demonstrates that low Rnd3 expression in lung adenocarcinoma is associated with longer survival.

## Knocking down Rnd3 does not affect cell viability or cell cycle progression in lung adenocarcinoma cell lines

To evaluate loss of Rnd3 expression on cell proliferation and cell death, we used two lung adenocarcinoma cell lines, A549 and H460. We knocked down Rnd3 expression using two separate siRNA oligonucleotides against Rnd3 (Rnd3(A) and Rnd3(D)) and non-silencing control (NSC) oligos for control, then performed Cell Counting Kit-8 (CCK8) assays at 24, 48, and 96 h post-knockdown. Knockdown of Rnd3 expression was confirmed at each time-point (Figs 2A and S2A and B); however, we observed no significant changes in cell number compared with NSC cells at each time-point in either A549 or H460 cell lines (Fig 2B and C).

Altered Rnd3 expression has been reported to affect cell cycle progression in several cell types (Villalonga et al, 2004; Bektic et al,

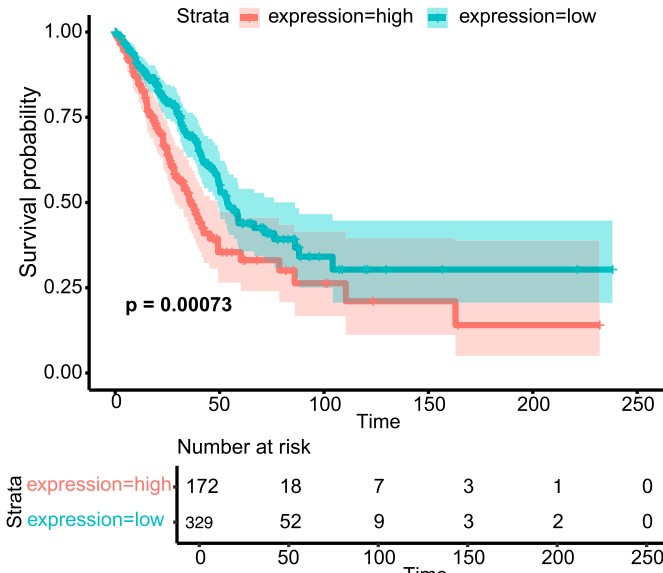

**Figure 1. Lung adenocarcinoma patients expressing low levels of Rnd3 have significantly higher survival probability rates, using optimal cutoff groupings.**
Kaplan–Meier plots of transcript data from TCGA-LUAD using optimal cutoff groupings of Rnd3 expression, comparing survival probability rates of lung adenocarcinoma patients with high Rnd3 (n = 172/501) versus low Rnd3 (n = 329/501) expression, $P$ = 0.00073, with risk table.

2005; Poch et al, 2007; Luo et al, 2012; Tang et al, 2014, 2018; Zhu et al, 2014; Clarke et al, 2015; Liu et al, 2015; Ma et al, 2020). To further confirm our cell counting data, cell proliferation was analyzed with propidium iodide staining and flow cytometry after 48 h of Rnd3 knockdown in both A549 and H460 cells (Figs 2A and S2A and B). We observed no effect on cell cycle progression after Rnd3 knockdown in A549 or H460 cells in either the G1 or the G2/M phase (Fig 2D). Taken together, these data indicate that loss of Rnd3 expression in lung adenocarcinoma cell lines does not alter cell viability or cell proliferation.

## Knocking down Rnd3 decreases cell invasion and migration in lung adenocarcinoma cell lines

As metastasis is an important factor in patient survival, we investigated whether the metastatic potential (cell invasion and migration) was altered in lung adenocarcinoma cells after the depletion of Rnd3 expression. To test this, we first performed invasion assays with A549 and H460 cells after Rnd3 knockdown. Rnd3 was depleted with siRNA for 24 h; cells were then seeded onto Matrigel-coated transwell inserts with 8-$\mu$m pores and allowed to invade for 24 h (A549) or 48 h (H460). These time-points were chosen because of differing rates of basal invasion for each cell line and allow for ~50% invasion of each cell lines' control at the assay endpoint. Transwells were fixed, stained, and imaged, and the invading cells were counted and normalized to the number of invading NSC cells. Rnd3 knockdown was confirmed via Western blot analysis (Fig 3A–C). We observed a significant decrease ($P$ < 0.0001) in invasion in both Rnd3-depleted A549 and H460 cells compared with NSC cells (Fig 3D–F).

We next investigated the effect of knocking down Rnd3 on cell migration by performing wound healing assays. Rnd3 was knocked down with siRNA in A549 and H460 cells for 48 h before a "wound" was introduced to the confluent monolayer of cells. The wound was imaged at time zero and at various intervals until the endpoints of 24 h for A549 cells and 48 h for H460 cells; at each endpoint, cells were lysed and Rnd3 knockdown was confirmed. Because of differing rates of basal migration for A549 and H460 cells, time-points of 24 and 48 h were chosen, which result in ~60% closure of control cells. After Rnd3 depletion, both A549 and H460 cells migrated significantly slower than their NSC control counterparts (Fig 3G–K). These data indicate that knocking down Rnd3 expression in two widely used lung adenocarcinoma cell lines decreases the cells' ability to both invade and migrate; these data are consistent with other studies in other cell lines.

## ROCK1 inhibition does not rescue the decreased cell invasion or migration phenotype of Rnd3-depleted lung adenocarcinoma cell lines

Rnd3 alters the actin cytoskeleton (Riento et al, 2003, 2005a; Wennerberg et al, 2003) and cell migration (Guasch et al, 1998; Klein et al, 2008; Pinner & Sahai, 2008; Klein & Aplin, 2009) via the RhoA-ROCK1 signaling pathway in several cell types. Specifically, Rnd3 inhibits RhoA and ROCK1 signaling, resulting in the turnover of actin stress fibers and focal adhesions allowing for a steady rate of cell migration (Guasch et al, 1998; Wennerberg et al, 2003; Pinner & Sahai, 2008). For example, in melanoma cells depletion of Rnd3 results in ROCK1 activation and increases the formation of actin stress fibers and focal adhesions, which anchor the cells to the extracellular matrix and decrease migration and invasive outgrowth. The ROCK1/2 inhibitor Y-27632 rescues the Rnd3-dependent migration and invasion phenotype in melanoma cells by restoring migration and invasion rates back to WT (Klein et al, 2008; Klein & Aplin, 2009). The ROCK1/2 inhibitor Y-27632 is commonly used to inhibit ROCK1 activity; however, at the common working concentration of 5 $\mu$M, Y-27632 also inhibits ROCK2 and PRK2 activity (Davies et al, 2000; Ishizaki et al, 2000). To determine whether Rnd3 signals via the RhoA-ROCK1 pathway in lung adenocarcinoma cells, we treated A549 and H460 cells with Y-27632 (Ishizaki et al, 2000) to test whether inhibiting ROCK1 rescued the effect of Rnd3 depletion on cell invasion and migration. First, we treated NSC and Rnd3-knockdown A549 and H460 cells with either 5 $\mu$M Y-27632 (+Y) or DMSO (−Y) as a control for 24 h; Y-27632 had no effect on Rnd3 expression (Figs 4K and S3A and B). We did observe the characteristic effect of inhibiting ROCK1 on the actin cytoskeleton; treatment with 5 $\mu$M Y-27632 significantly decreased actin stress fibers, to almost complete ablation, in both NSC and Rnd3-knockdown cells compared with DMSO control cells (Fig 4A–D), indicative of ROCK1 inhibition. Increasing the concentration of Y-27632 to 10 $\mu$M did not result in a further loss of actin stress fibers (Fig S3C–F).

Next, we investigated the effect ROCK1 inhibition had on A549 and H460 cell invasion and migration. Treatment with either 5 or 10 $\mu$M of the ROCK1/2 inhibitor Y-27632 had no effect on invasion or migration rate in WT or NSC A549 cells (Figs 4E, G, I, and J and S3G). Treatment with 5 $\mu$M Y-27632 did significantly increase

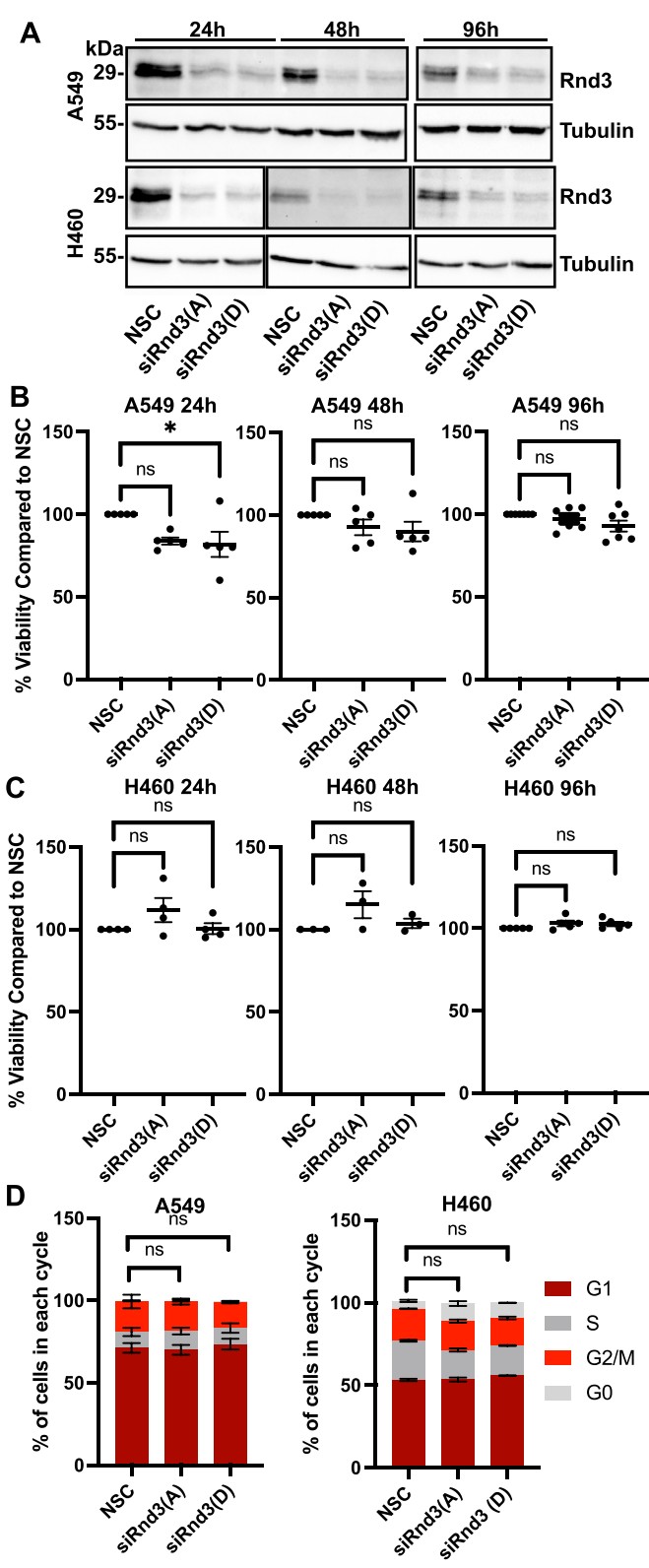

**Figure 2. Knockdown of Rnd3 expression in A549 or H460 cells does not alter cell viability or cell cycle progression.**
**(A)** Representative Western blots in which decreased Rnd3 expression is observed in A549 and H460 cells transfected with two separate Rnd3 siRNA oligos A (Rnd3(A)) and D (Rnd3(D)) compared with non-silencing control cells at 24, 48,

(*P* = 0.0103) the invasion rate of NSC H460 cells compared with DMSO-treated NSC H460 cells (Fig 4F and H), and no additional increase in invasion was observed after 10 μM Y-27632 treatment (Fig S3H). These data indicate that 5 μM Y-27632 treatment inhibits ROCK1 signaling, as ablation of actin stress fibers was observed, as well as increased invasion rate of H460 control cells. However, treatment with 5 μM Y-27632 did not rescue the invasion or migration rate of either A549 or H460 Rnd3-knockdown cells back to WT levels (Fig 4E–J). No significant difference in invasion rates was observed in Rnd3-knockdown cells treated with 5 μM Y-27632 compared with DMSO-treated Rnd3-knockdown cells, for either Rnd3 oligo or cell line (A549 siRnd3 oligo A, p ≥ 0.999, oligo D, *P* = 0.761; H460 siRnd3 oligo A, *P* = 0.998, oligo D, *P* = 0.995) (Fig 4E and F). The same pattern was observed for cell migration in Rnd3-knockdown A549 cells treated with either DMSO or Y-27632; however, no significant effect on the cell migration rate was observed (*P* = 0.640) (Fig 4J). These data indicate that Y-27632 treatment inhibits ROCK1 signaling as we observed complete loss of actin stress fibers; however, ROCK1 inhibition with Y-27632 could not rescue migration or invasion rates of Rnd3-knockdown A549 or H460 cells.

In addition, we did not observe an increase in actin stress fibers after Rnd3 knockdown in either A549 or H460 cells (Fig 4A–D), as reported in other cell lines (Klein et al, 2008; Klein & Aplin, 2009). We observed the number of actin stress fibers significantly decreased in Rnd3-knockdown A549 cells, with no significant change to actin stress fiber number in H460 Rnd3-knockdown cells compared with NSC cells (Fig 4A–D). We also did not observe an increase in the phosphorylation of ROCK1 downstream targets, MLC2 or Cofilin, indicating knocking down Rnd3 in these lung adenocarcinoma cells does not lead to increased ROCK1 signaling or an increase in actin stress fiber number (Figs 4A–D and S4A–I). Taken together, these data indicate that loss of Rnd3 expression did not induce a significant increase in ROCK1 signaling and that inhibiting ROCK1 with Y-27632 did not restore cell migration or invasion rates in Rnd3-knockdown A549 or H460 cells.

### Specific knockdown of ROCK1 or ROCK2 does not rescue the invasion or migration rates of Rnd3-depleted lung adenocarcinoma cells

The ROCK1/2 inhibitor Y-27632 is commonly used to inhibit ROCK1 activity. However, at the common working concentration of 5 μM, Y-27632 also inhibits ROCK2 and PRK2 activity (Davies et al,

and 96 h post-transfection. Tubulin was used as a loading control, n = 4 (24 h), n = 5 (48 h), and n = 6 (96 h). **(B, C)** Percentage of cell viability compared with NSC cells at 24, 48, and 96 h post-transfection in (B) A549 and (C) H460 cells. Data are presented as an individual mean for each experiment with bar representing overall mean ± SEM. Statistical comparisons were performed using one-way ANOVA with Dunnett's multiple test correction. **(D)** Flow cytometry of propidium iodine incorporation was performed 48 h post-transfection with Rnd3 siRNA oligos A (Rnd3(A)) and D (Rnd3(D)) to deplete Rnd3 expression compared with non-silencing control in A549 cells (n = 5) and H460 cells (n = 3). Cell cycle statistical comparisons were performed for G1 and G2/M comparison using one-way ANOVA with Dunnett's multiple comparison test. *P* < 0.05, ns, not significant.

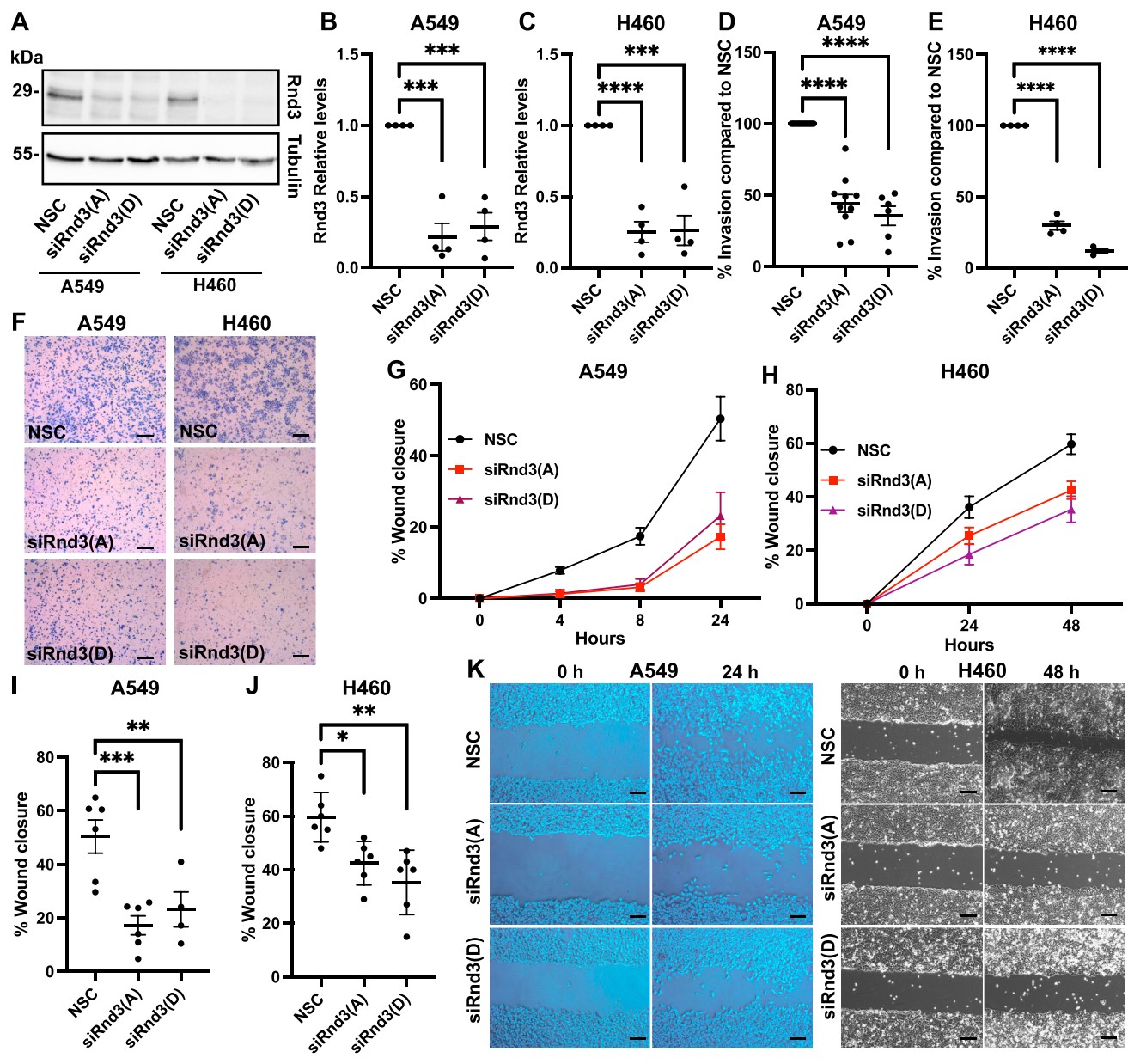

**Figure 3. Knockdown of Rnd3 expression in lung adenocarcinoma cell lines decreases cell invasion and cell migration.**
**(A, B, C)** Decreased Rnd3 expression is observed in A549 and H460 cells transfected with Rnd3 siRNA oligos A (Rnd3(A)) and D (Rnd3(D)) compared with non-silencing control (NSC) cells. Tubulin was used as a loading control. **(A, B, C)** Representative Western blots and (B, C) densitometric analysis were performed for each sample set; the expression level of Rnd3 was normalized to own tubulin loading control, then normalized to the expression level of Rnd3 in the NSC cells in (B) A549 and (C) H460 cells, n = 4. **(D, E, F)** Invasion assay across a transwell membrane coated with Matrigel. **(D, E)** Percentage of invaded cells compared with NSC cells at (D) 24 h for A549 cells (n = 6) and (E) 48 h for H460 cells (n = 4). Data are presented as an individual mean for each experiment, and normalized to own NSC, with bar representing overall mean ± SEM. **(F)** Representative images of invading cells at 24 h (A549) and 48 h (H460), 10× magnification; scale bar = 100 $\mu$m. **(G, H, I, J, K)** Wound healing assays. **(G, H)** Time course, percentage wound closure at each time-point compared with wound at 0 h, for (G) A549 cells and (H) H460 cells; data are presented as mean ± SEM. **(I, J)** Percentage wound closure at (I) 24 h (A549 cells, n = 6) and (J) 48 h (H460 cells, n = 6) compared with wound at 0 h. Data are presented as an individual mean for each experiment with bar representing overall mean ± SEM. **(K)** Representative images of wound closure at 0 h and either 24 h for A549 or 48 h for H460 cells, 10× magnification; scale bar = 100 $\mu$m. Statistical comparisons were performed using one-way ANOVA with Dunnett's multiple test correction. *$P < 0.05$, **$P < 0.01$, ***$P < 0.001$, ****$P < 0.0001$.

2000; Ishizaki et al, 2000). ROCK1 and ROCK2 share 64% homology but have distinct and sometimes opposing roles (Lock et al, 2012; Julian & Olson, 2014). To specifically target ROCK1, we knocked down ROCK1 to ascertain whether this could rescue invasion and migration rates back to WT in Rnd3-knockdown lung adenocarcinoma cells (Fig 5). As a proof-of-concept experiment, we first tested this approach in the melanoma cell line, A375, in which Klein et al have shown knocking down Rnd3 expression decreases cell migration in a ROCK1-dependent manner (Klein et al, 2008; Klein & Aplin, 2009; Katiyar & Aplin, 2011). We performed ROCK1- and ROCK2-specific

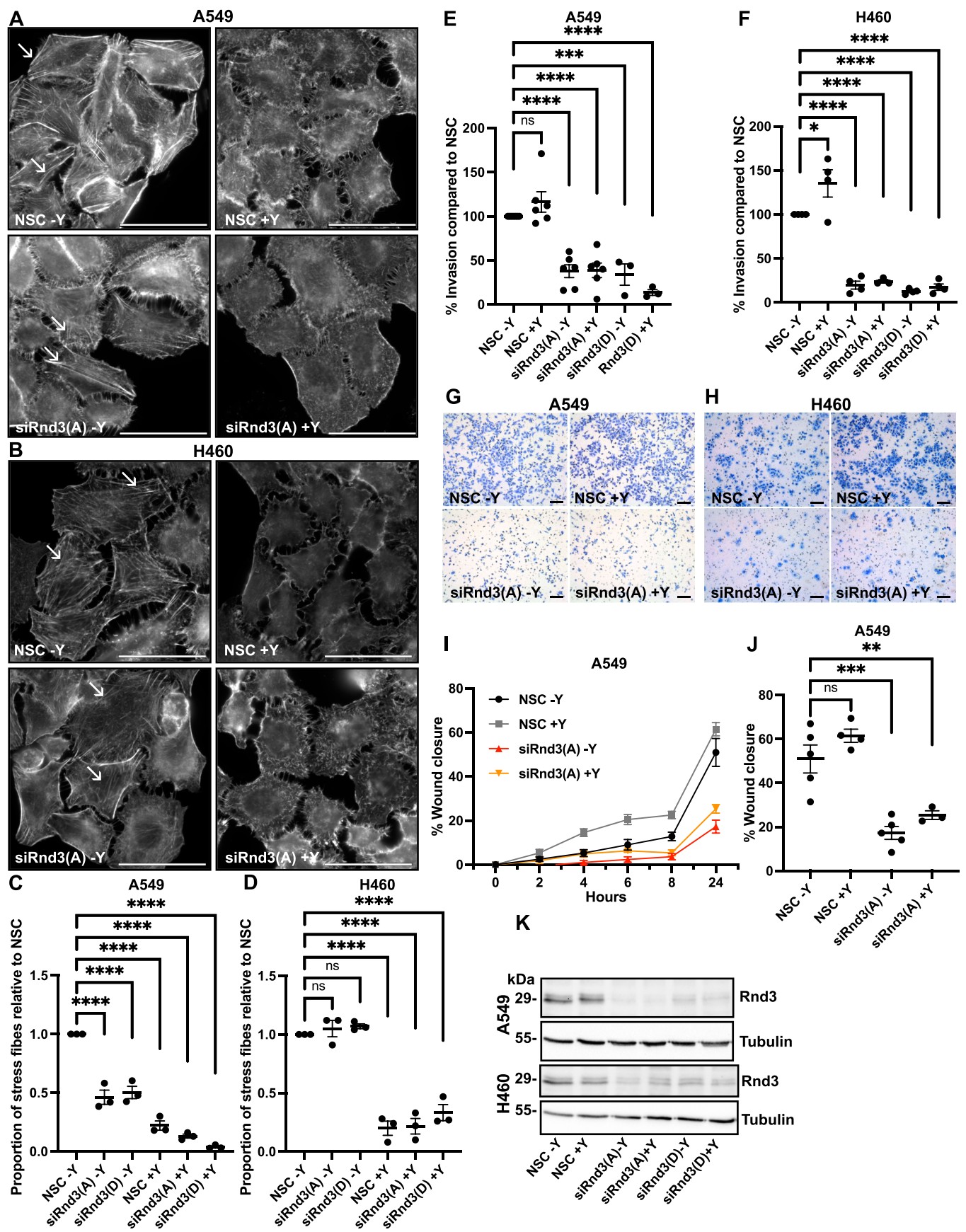

siRNA knockdown, as well as Rnd3:ROCK1 and Rnd3:ROCK2 double knockdowns in the A375 melanoma cell line (Figs 5C and S5G–I). We observed that knocking down Rnd3 in the melanoma cells significantly decreased cell invasion, and that either ROCK1 or ROCK2 double knockdown with Rnd3 rescued invasion rates back to levels not significantly different from NSC cells (NSC compared with siRnd3:ROCK1, $P = 0.384$ and siRnd3:ROCK2, $P = 0.702$) (Fig 5F). These data indicate that Rnd3 signals through ROCK1 and ROCK2 in melanoma cells to control migration and invasion, which is in line with published data (Klein et al, 2008; Klein & Aplin, 2009; Katiyar & Aplin, 2011). We then repeated this experiment in A549 and H460 lung adenocarcinoma cells, significantly knocking down ROCK1 or ROCK2 separately, as well as Rnd3:ROCK1 and Rnd3:ROCK2 double knockdowns (Figs 5A and B and S5A–F). Specific knockdown of ROCK1 or ROCK2 had no significant effect on A549 invasion or migration and did not rescue invasion or migration rates in Rnd3-knockdown A549 cells as both invasion and migration rates remained significantly reduced compared with control cells (Fig 5D, G, and H). In addition, when comparing the invasion or migration rates of Rnd3-knockdown cells to siRnd3:ROCK1 or Rnd3:ROCK2 double knockdown in A549 cells, no significant change in invasion rate (siRnd3 compared with siRnd3:ROCK1, $P = 0.972$, or siRnd3:ROCK2, $P = 0.963$) or migration rate was observed (siRnd3 compared with siRnd3:ROCK1, $P = 0.681$, or siRnd3:ROCK2, $P = 0.988$). Knocking down ROCK1 or ROCK2 in H460 cells resulted in a significant increase in cell invasion compared with NSC cells; however, double knockdowns of ROCK1 or ROCK2 with Rnd3 did partially rescue the invasion rates back to WT in the Rnd3-depleted H460 cells (Fig 5E). However, when comparing the invasion rates of Rnd3 knockdown to siRnd3:ROCK1 or Rnd3:ROCK2 double knockdown in H460 cells, no significant change in invasion rate was observed (siRnd3 compared with siRnd3:ROCK1, $P = 0.910$, or siRnd3:ROCK2, $P = 0.237$). These data suggest, along with the comparison with NSC cells, that knocking down Rnd3 expression in lung adenocarcinoma cell lines, A549 and H460, decreases both cell invasion and migration via a ROCK1/2-independent pathway.

### Rnd3 signaling maintains control over cell invasion in patient-derived lung-to-brain metastasis (PDLBM) cell lines

With lung-to-brain metastasis being the leading cause of mortality in lung cancer patients (Chi & Komaki, 2010; Yousefi et al, 2017), we investigated whether Rnd3 signaling continues to promote metastatic potential in advanced metastatic disease. Two patient-derived brain metastatic non–small-cell lung carcinoma cells (PDLBM 601 and 620) were used as a proxy for advanced disseminated disease. Tumors were surgically removed from patients, and cells from these tumors were maintained in culture. We knocked down Rnd3 expression in both PDLBM cell lines, 601 and 620, and observed a significant reduction in cell invasion compared with NSC cells (Figs 6A–D and S6A and B). We also investigated whether Rnd3's regulation of cell invasion in the PDLBM cells was dependent on ROCK1 signaling. We were unable to rescue cell invasion back to WT rates in the Rnd3:ROCK1 double knockdowns in either 601 or 620 PDLBM cell lines (Figs 6E–G and S6C–F). In addition, when comparing the invasion rates of Rnd3 knockdown to siRnd3:ROCK1 double knockdown in 601 cells, no significant change in invasion rate was observed (siRnd3 compared with siRnd3:ROCK1, $P = 0.899$). Similarly, when comparing the invasion rates of Rnd3 knockdown to siRnd3:ROCK1 double knockdown in 620 cells, no significant change in invasion rate was observed (siRnd3 compared with siRnd3:ROCK1, $P = 0.997$). These data indicate, along with the comparison with NSC cells, that Rnd3 mediates signaling pathways, which are independent of ROCK1 signaling, to control metastatic potential in advanced lung-to-brain metastatic disease.

### Knocking down RhoA expression does not rescue Rnd3 migration or invasion rates in lung adenocarcinoma cells

We next investigated the potential signaling pathways in which Rnd3 may be regulating metastatic potential in lung adenocarcinoma cells. We first investigated RhoA signaling, a known mediator of the actin cytoskeleton and cell migration (Ridley, 2013; Schaks et al, 2019). Rnd3 has been shown to inhibit RhoA activity by recruiting p190RhoGAP to RhoA and inactivating RhoA (Wennerberg et al, 2003). Although RhoA commonly signals via ROCK1 to regulate the actin cytoskeleton and cell migration, additional RhoA-mediated signaling pathways affect cell migration and invasion (Ridley, 2013; Schaks et al, 2019). We performed double Rnd3:RhoA-knockdown experiments, using two separate siRNA oligos against RhoA (oligo RhoA(1) and RhoA(2)), and assayed cell migration in A549 cells and cell invasion in H460 cells (Fig 7). We observed a significant decrease in cell migration after single RhoA knockdowns, indicating loss of RhoA expression decreases cell migration in A549 cells. However, knocking down RhoA in Rnd3-knocked down A549 cells was unable to rescue cell migration rates back to WT

**Figure 4. Inhibition of ROCK1 does not rescue the invasion or migration phenotype of Rnd3 knockdown in lung adenocarcinoma cells.**
**(A, B, C, D)** 24-h treatment with 5 $\mu$M Y-27632 (+Y) induces the loss of actin stress fibers compared with DMSO control (–Y) in (A, C) A549 and (B, D) H460 cells. **(A, B)** Representative epifluorescence images of the actin stain phalloidin (white). White arrows highlight actin stress fibers, 60× magnification; scale bar = 50 $\mu$m. **(C, D)** Quantification of actin stress fibers; five fields of view were imaged using a 60× oil objective, a total number of actin stress fibers per cell in five cells per field of view were counted, and the average actin stress fiber number per cell was calculated and normalized to own DMSO-treated NSC for each condition per biological repeat, n = 3. **(E, F, G, H)** Invasion assay across a transwell membrane coated with Matrigel, treated with 5 $\mu$M Y-27632 (+Y) or DMSO as a control (–Y). **(E, F)** Percentage of invaded cells compared with DMSO-treated NSC (–Y) cells at (E) 24 h for A549 and (F) 48 h for H460 cells, n = >4; data are presented as mean ± SEM. **(G, H)** Representative images of invading cells at (G) 24 h for A549 and (H) 48 h for H460 cells, 10× magnification; scale bar = 100 $\mu$m. **(I, J)** Wound healing assay, treated with 5 $\mu$M Y-27632 (+Y) or DMSO control (–Y). **(I)** Time course, percentage wound closure for A549 cells at each time-point compared with wound at 0 h, and data are presented as mean ± SEM. **(J)** Percentage wound closure for A549 cells at 24 h compared with wound at 0 h, n = 4. Data are presented as an individual mean for each experiment with bar representing overall mean ± SEM; statistical comparisons were performed using one-way ANOVA with Dunnett's multiple test correction. *$P < 0.05$, **$P < 0.01$, ***$P < 0.001$, ****$P < 0.0001$, ns, not significant. **(K)** Representative Western blots of samples used in above experiment, in which decreased Rnd3 expression is observed in A549 and H460 cells transfected with siRNA oligos A (Rnd3(A)) and D (Rnd3(D)), treated with 5 $\mu$M Y-27632 (+Y) or DMSO (–Y). Tubulin was used as a loading control.

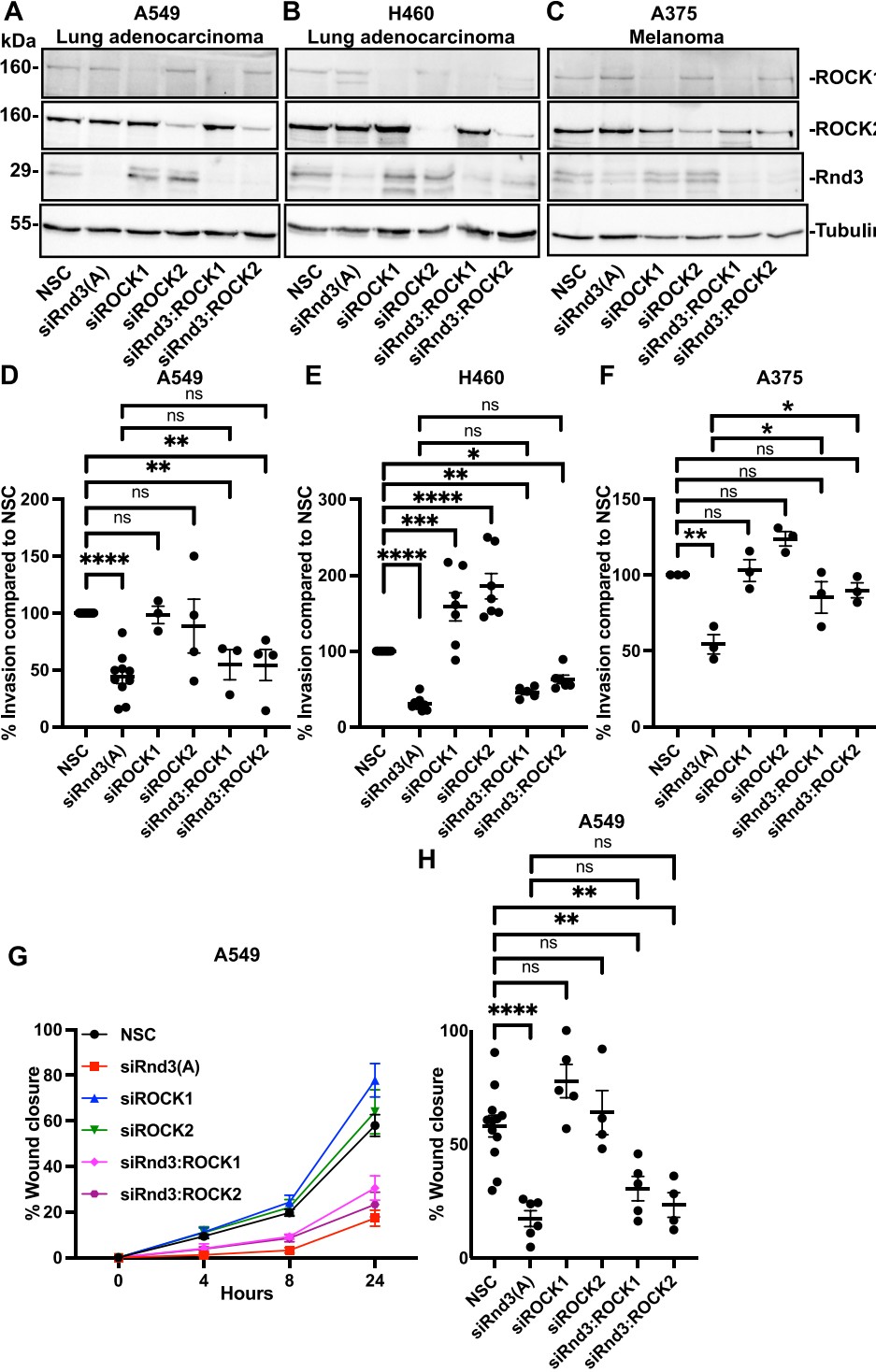

**Figure 5. Knockdown of ROCK1 or ROCK2 does not fully rescue the migration or invasion phenotype of Rnd3 knockdown in A549 or H460 lung adenocarcinoma cells.**
**(A, B, C)** Representative Western blots in which decreased protein expression levels of Rnd3 (oligo A), ROCK1, and ROCK2 are observed, corresponding to siRNA treatment compared with non-silencing control (NSC) in (A) A549, (B) H460, and (C) A375 cells (as an experimental control). Tubulin was used as a loading control. **(D, E, F)** Invasion assay across a transwell membrane coated with Matrigel, percentage of invaded cells compared with NSC cells at (D) 24 h for A549, n = >4, (E) 48 h for H460, n = 6, and (F) 16 h for A375 cells, n = 3 (as an experimental control). **(G, H)** A549 wound healing assay. **(G)** Time course, percentage wound closure at each time-point compared with wound at 0 h, and data are presented as mean ± SEM. **(H)** Percentage wound closure at 24 h compared with wound at 0 h, n = 5. Data are presented as an individual mean for each experiment with bar representing overall mean ± SEM. Statistical comparisons were performed using one-way ANOVA with Dunnett's multiple test correction. *$P < 0.05$, **$P < 0.01$, ***$P < 0.001$, ****$P < 0.0001$, ns, not significant.

(Fig 7A, C–E). We also performed invasion assays with H460 cells with single and double Rnd3 and RhoA knockdowns (Fig 7C, F–G). Knocking down RhoA in H460 cells significantly decreased their ability to invade, and like the A549 migration data, double knockdown of Rnd3:RhoA did not rescue the invasion phenotype back to WT levels (Fig 7B). In addition, when comparing Rnd3 single knockdowns to Rnd3:RhoA double knockdowns, no significant difference was observed in either cell migration rates in A549 cells

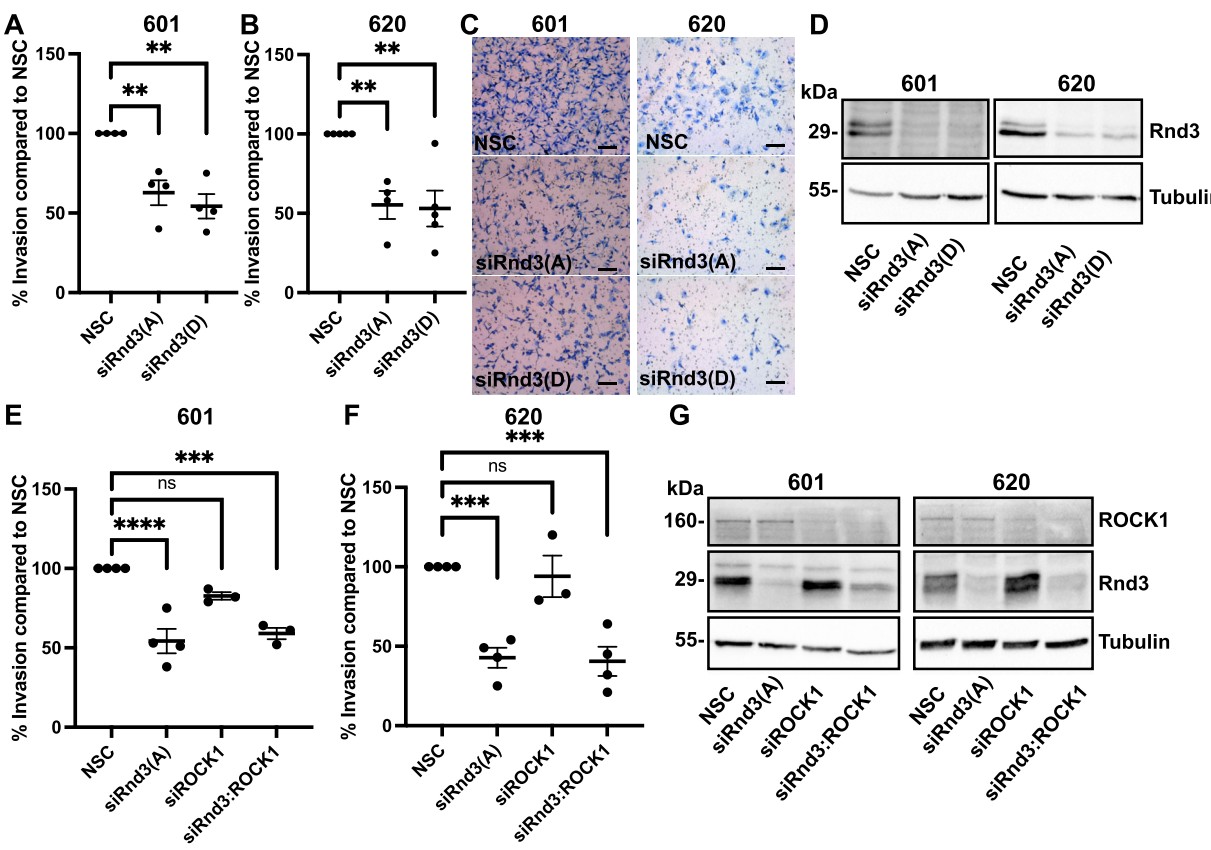

**Figure 6. Knockdown of Rnd3 expression in patient-derived lung-to-brain metastasis cell lines decreases cell invasion independently of ROCK1 signaling.**
**(A, B, C, E, F)** Invasion assay across a transwell membrane coated with Matrigel. **(A, B, E, F)** Percentage of invaded cells compared with NSC cells at 24 h for (A, E) 601 and (B, F) 620 cells, n = 4. **(C)** Representative images of invading cells at 24 h, 10x magnification; scale bar = 100 $\mu$m. **(D, G)** Representative Western blots in which decreased protein expression levels of Rnd3 expression are observed in 601 and 620 cells transfected with Rnd3 siRNA oligos A (Rnd3(A)) and D (Rnd3(D)) (D) and decreased protein expression levels of Rnd3 and ROCK1 are observed (G), corresponding to siRNA treatment compared with non-silencing control cells. Tubulin was used as a loading control. Data are presented as an individual mean for each experiment with bar representing overall mean ± SEM; statistical comparisons were performed using one-way ANOVA with Dunnett's multiple test correction. **\*\***$P < 0.01$, **\*\*\***$P < 0.001$, **\*\*\*\***$P < 0.0001$, ns, not significant.

(siRnd3 compared with siRnd3:RhoA(1), $P$ = 0.999, or siRnd3:RhoA(2), $P$ = 0.710) (Fig 7A) or cell invasion rates for H460 cells (siRnd3 compared with siRnd3:RhoA(1), $P$ = 0.327, or siRnd3:RhoA(2), $P$ = >0.999) (Fig 7B). These data indicate RhoA expression is required for cell migration and invasion to occur and that loss of RhoA is unable to restore the migration or invasion phenotype of Rnd3-deleted A549 or H460 cells, respectively.

### Rnd3 regulates cell invasion and migration via alpha 5 integrin expression in A549 lung adenocarcinoma cells

Cell adhesion to the extracellular matrix plays an important role in cell migration and invasion (Baster et al, 2025). We investigated whether the expression levels of the adhesion molecules, integrins, were altered in Rnd3-depleted A549 cells. Out of a panel of integrins: alpha 5, alpha V, beta 1, beta 3, and beta 5, we observed no change in the expression levels of alpha V, beta 1, or beta 5 (Fig S7A, B, D–I). However, we did observe a significant decrease in beta 3 expression and a significant increase in alpha 5 integrin expression in Rnd3-knocked down A549 cells compared with NSC cells (Figs S7C, J, and K and S8A–C). In addition to increased alpha

5 protein expression observed via Western blotting analysis, we also observed a significant increase in fluorescence intensity of alpha 5 staining in Rnd3-knocked down A549 cells compared with NSC cells (Figs 8A and B and S8D). Alpha 5 staining was ablated in both control and Rnd3-knockdown A549 cells after alpha 5 siRNA knockdown (Fig 8A and B). We next investigated whether the increase in alpha 5 expression could affect the cells' ability to invade and/or migrate. We performed single and double Rnd3:Alpha 5–knockdown experiments, using two separate siRNA oligos against alpha 5 (oligo Alpha 5[A] and Alpha 5[B]) (Figs 8E–G and S8E–G), and assayed cell invasion (Fig 8C) and cell migration (Fig 8D) in A549 cells. Knocking down alpha 5 integrin expression alone had no significant effect on the invasion rate of A549 cells; however, double knockdown of alpha 5 integrin and Rnd3 was able to restore A549 cells' ability to invade (Fig 8C). Similarly, depleting alpha 5 expression did not affect migration rates of A549 cells but did rescue migration rates of Rnd3-knocked down A549 cells (Fig 8D). These data indicate a novel mechanism whereby Rnd3 is regulating cell migration and invasion in lung adenocarcinoma cells via modulating alpha 5 integrin expression.

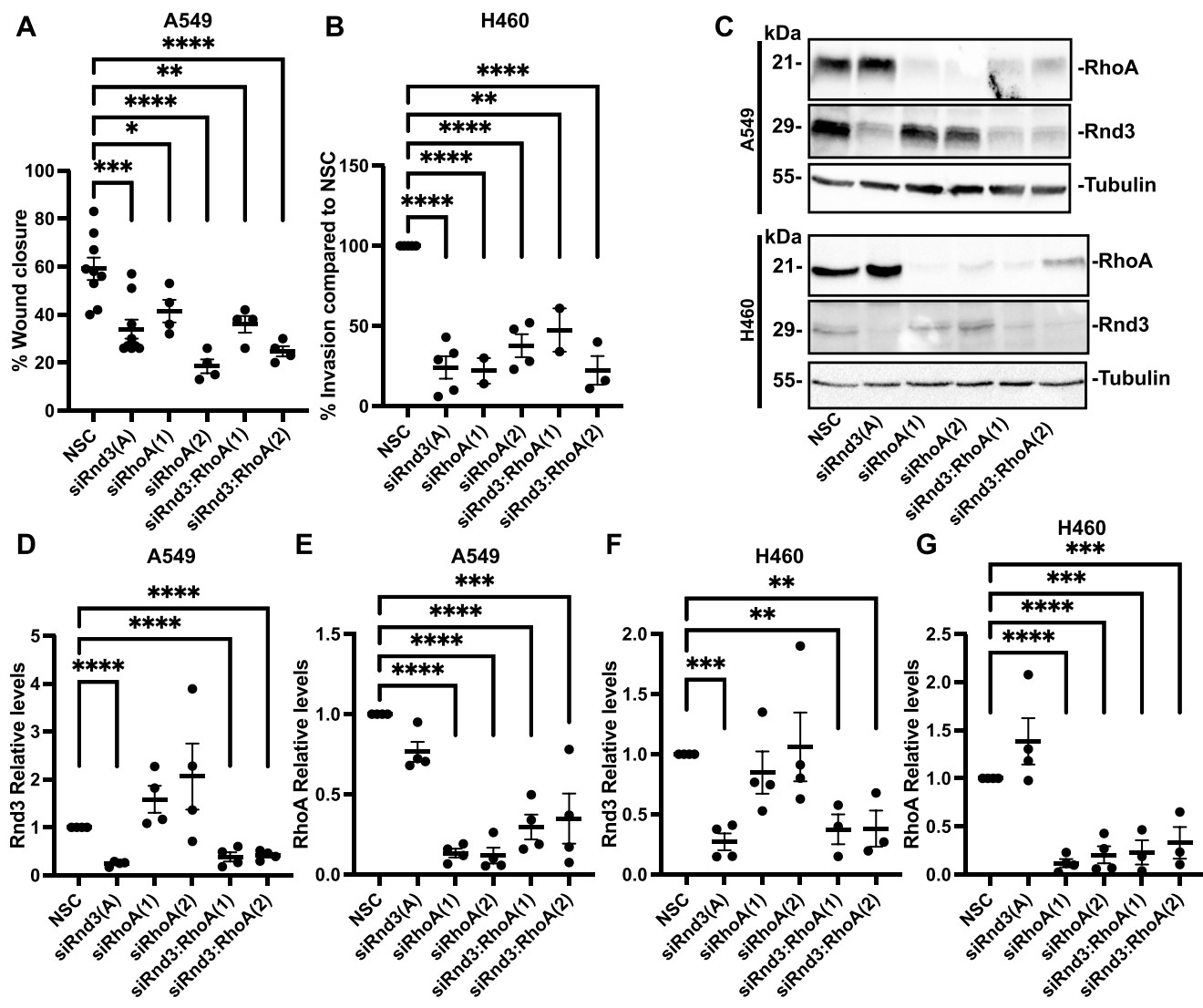

**Figure 7. Knockdown of RhoA does not rescue the migration or invasion phenotype of Rnd3 knockdown in lung adenocarcinoma cells.**
**(A)** Wound healing assay using A549 cells, and percentage wound closure at 24 h compared with wound at 0 h. **(B)** Invasion assay across a transwell membrane coated with Matrigel, and percentage of invaded cells compared with NSC H460 cells at 48 h. **(C)** Representative Western blots in which decreased protein expression levels of Rnd3 (oligo A) and RhoA (oligos 1 & 2) are observed, corresponding to siRNA treatment compared with non-silencing control, A549, and H460 cells. Tubulin was used as a loading control. **(D, E, F, G)** Quantification of representative Western blots. Densitometric analysis was performed for each sample set over multiple biological replicates for Rnd3 (D, F) and RhoA (E, G) expression after transient transfection with siRNA oligo compared with expression levels in NSC cells in (D, E) A549 and (F, G) H460 cells, n = 4. Expression levels of each protein of interest were normalized to own tubulin loading control, then normalized to the expression level of that protein of interest in the NSC cells. Data are presented as an individual mean for each experiment with bar representing overall mean ± SEM. Statistical comparisons were performed using one-way ANOVA with Dunnett's multiple test correction. *$P < 0.05$, **$P < 0.01$, ***$P < 0.001$, ****$P < 0.0001$.

## Discussion

Lung adenocarcinoma accounts for ~40–50% of all lung cancers (Travis et al, 2011; Herbst et al, 2018; Duma et al, 2019). Most lung cancers metastasize (Siegel et al, 2021), with lung-to-brain metastasis being the leading cause of mortality in lung cancer patients (Chi & Komaki, 2010; Yousefi et al, 2017). The identification of key regulators and new therapeutic targets in lung cancer metastasis is still an area of active research. Expression levels of Rnd3, rather than mutations in Rnd3, have been linked to cancer. In lung cancer, a variety of methods, including immunohistochemistry and qRT-PCR, have been used to show that Rnd3 is overexpressed

compared with normal tissue, and overall survival for lung cancer patients with high Rnd3 expression was lower than for those with low Rnd3 expression (from several different patient cohorts) (Zhang et al, 2007; Raz et al, 2008; Li et al, 2016; Sun et al, 2016). Our specific analysis of TCGA-LUAD (The Cancer Genome Atlas Lung Adenocarcinoma) data supports these findings, showing that lung adenocarcinoma patients expressing high levels of Rnd3 have significantly lower survival probability, using both optimal and top and bottom quartile cutoffs. Together, these data further support the proposal that elevated Rnd3 expression is detrimental to the survival of patients with lung cancer. As such, Rnd3 has the potential to be an independent marker for lung cancer–related

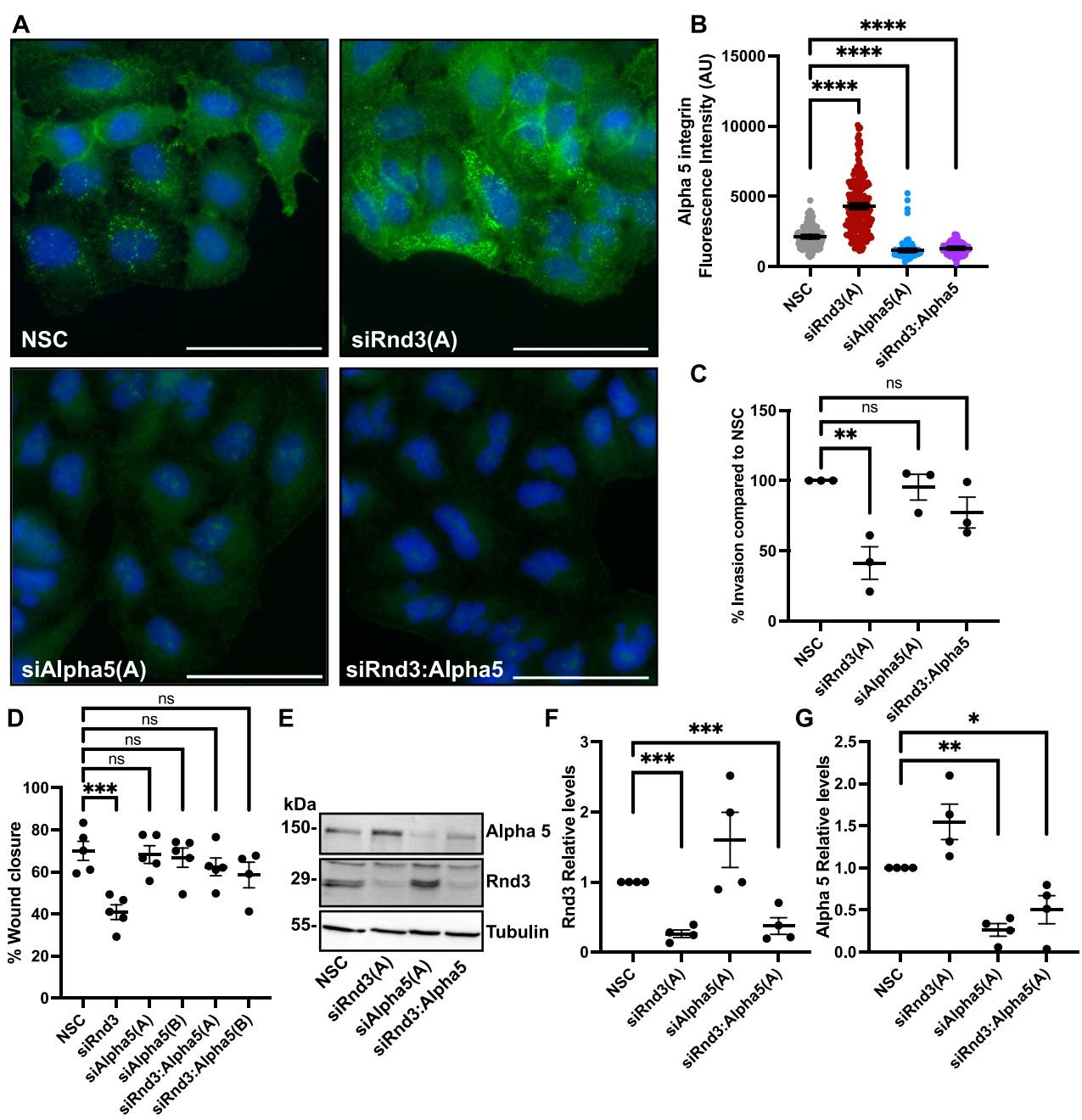

Figure 8. **Knocking down alpha 5 integrin expression restores the invasion and migration rates of Rnd3-knocked down A549 cells.**
**(A)** Representative epifluorescence images of anti-alpha 5 integrin (green) and nuclear DAPI stain (blue) in A549 NSC and siRnd3 (oligo A), siAlpha5 (oligo A), and siRnd3:Alpha5–knockdown A549 cells, imaged with equal exposure times using a 60x oil objective; scale bar = 50 $\mu$m. **(B)** Analysis of alpha 5 fluorescence intensity (per biological repeat, five fields of view were captured with equal exposure time using a 40x objective per condition, 10 cells plus a section of background were defined as region of interest in Nikon NIS-Elements software, and the average intensity for each cell minus background was determined). Data are presented as individual cells with bar representing overall mean for each biological repeat ± SEM, n = 4. **(C)** Invasion assay across a transwell membrane coated with Matrigel, and percentage of invaded cells compared with NSC cells at 24 h. Data are presented as an individual mean for each experiment with bar representing overall mean ± SEM, n = 3. **(D)** Wound healing assay. Percentage wound closure at 24 h compared with wound at 0 h. Data are presented as an individual mean for each experiment with bar representing overall mean ± SEM, n = 4. **(E)** Representative Western blots in which decreased protein expression levels of Rnd3 (oligo A) and alpha 5 (Alpha 5) integrin (oligo A) are observed, corresponding to siRNA treatment compared with the non-silencing control. Tubulin was used as a loading control. **(F, G)** Quantification of representative Western blots. **(F, G)** Densitometric analysis was performed for each sample set over multiple biological replicates, n = 4, for Rnd3 (F) and alpha 5 (G) expression after transient transfection with siRNA oligo compared with expression levels in NSC A549 cells. Expression levels of each protein of interest were normalized to own tubulin loading control, then normalized to the expression level of that protein of interest in the NSC cells. Data are presented as an individual mean for each experiment with bar representing overall mean ± SEM. Statistical comparisons were performed using one-way ANOVA with Dunnett's multiple test correction. *$P$ < 0.05, **$P$ < 0.01, ***$P$ < 0.001, ****$P$ < 0.0001, ns, not significant; AU, arbitrary units.

survival and may serve as a molecular marker to identify high-risk individuals (Cuiyan et al, 2007; Zhang et al, 2007; Raz et al, 2008; Sun et al, 2016).

To investigate whether Rnd3 confers metastatic properties to lung cancer cells, we first investigated the effect of knocking down Rnd3 expression in lung adenocarcinoma as on cell proliferation. Several studies have reported that altered Rnd3 expression can affect cell proliferation. Both overexpression and decreased Rnd3 expression have been reported to affect cell cycle progression through a variety of different mechanisms, in different cell types and cancer types, with Rnd3's effect being potentially dependent on the mutational background of the cell. Other studies showed Rnd3 depletion to have no effect on melanoma cell proliferation (Klein & Aplin, 2009). Most of the studies investigating Rnd3's role in proliferation have not been performed in lung cancer cells (Villalonga et al, 2004; Bektic et al, 2005; Poch et al, 2007; Luo et al, 2012; Tang et al, 2014, 2018; Zhu et al, 2014; Clarke et al, 2015; Liu et al, 2015; Ma et al, 2020). For lung cancer cell lines, Tang et al reported that Rnd3 overexpression in H358 and H520 cells decreased proliferation (Tang et al, 2014), whereas depleting Rnd3 expression induced G1 arrest in A549 and H1299 cells (Tang et al, 2018). We observed that Rnd3 knockdown had no effect on the cell viability or cell cycle progression in A549 or H460 cells. Tang et al (2018) knocked down Rnd3 in A549 cells and reported decreased proliferation through G1 cell cycle arrest; however, only one siRNA oligo was used, and they did not report statistical significance (Tang et al, 2018). We used two independent siRNA oligonucleotides to knock down Rnd3 expression to mitigate off-target effects, and the knockdown was validated with Western blot. In addition, we did not observe an increase in cell death, as cell viability was unchanged in the Rnd3-knockdown cell lines compared with the control. Whether altering Rnd3 expression affects cell proliferation may depend on the background mutations present in different cell lines, which could bypass Rnd3's influence on cell proliferation and viability. Both A549 and H460 cells have KRAS mutations, which could drive cell proliferation and survival independently of Rnd3 expression levels. This does not rule out the possibility that in a stress situation, Rnd3-depleted cells could be more susceptible to cell death or cell cycle arrest. Taken together, our data suggest that Rnd3 knockdown in A549 or H460 lung adenocarcinoma cells does not affect cell cycle progression or cell viability, indicating Rnd3 is not regulating proliferation in lung adenocarcinoma cells.

We observed a significant decrease in both cell invasion and migration in A549 and H460 cells after knocking down Rnd3. These data corroborate other reports that Rnd3 depletion decreases cell invasion and migration in other cell types, including lung cancer, melanoma, and glioma (Klein & Aplin, 2009; Klein & Higgins, 2011; Clarke et al, 2015; Tang et al, 2018). Notably, cell invasion and migration are hallmarks of metastasis. The decrease in both invasion and migration observed in the Rnd3-depleted A549 and H460 cells could be an indicator of why lung cancer patients who have low Rnd3 expression survive longer. To determine whether Rnd3 maintains control over invasion in advanced disease, we used patient-derived lung-to-brain metastasis (PDLBM) cell lines. These cells were harvested from brain metastatic lesions of lung adenocarcinoma patients. We observed a significant decrease in

cell invasion in PDLBM cells after Rnd3 knockdown. These data strongly suggest that loss of Rnd3 expression decreases metastatic hallmarks (cell migration and invasion) and that this pathway is still functioning and can be manipulated in advanced disease, as depleting Rnd3 expression in the PDLBM cell lines decreased the cells' ability to invade. Understanding how Rnd3 is regulating these processes and potentially the metastatic potential of the cell would have far-reaching implications if we can target this pathway therapeutically.

We next investigated the mechanism of how Rnd3 regulates cell invasion and migration in lung adenocarcinoma cells. Rnd3 is known to interact with the RhoA-ROCK1 signaling pathway to regulate the actin cytoskeleton (Guasch et al, 1998; Riento et al, 2003, 2005a; Wennerberg et al, 2003; Klein et al, 2008; Pinner & Sahai, 2008; Klein & Aplin, 2009). Rnd3 inhibits RhoA and its downstream effector ROCK1, by recruiting p190RhoGAP to RhoA, as well as binding directly to ROCK1, inhibiting downstream signaling, resulting in the turnover of actin stress fibers and focal adhesions (Guasch et al, 1998; Wennerberg et al, 2003; Pinner & Sahai, 2008). Depletion of Rnd3 leads to ROCK1 activation and the formation of actin stress fibers and focal adhesions, which anchor cells to the extracellular matrix and decrease migration (Klein et al, 2008; Klein & Aplin, 2009). We observed the presence of actin stress fibers in both NSC and Rnd3-knockdown A549 and H460 cells, as well as the characteristic loss of actin stress fibers and altered cell shape after ROCK1 inhibition with Y-27632 treatment. Surprisingly, no increase in ROCK1 signaling was observed in either Rnd3-knocked down A549 or H460 cells, as we did not observe either an increase in actin stress fibers or increased phosphorylation of either direct or indirect ROCK1 downstream targets, MLC2 or Cofilin, compared with NSC cells. This was surprising because Rnd3 depletion has been shown to increase ROCK1 activity, increasing ROCK1 signaling and actin stress fiber formation (Guasch et al, 1998; Riento et al, 2003, 2005a; Klein et al, 2008; Klein & Aplin, 2009). These data indicate the following: (1) the presence of actin stress fibers, (2) ROCK1 is present and active, (3) Y-27632 was functioning to inhibit ROCK1 in both A549 and H460 cells, and (4) no increase in ROCK1 signaling was observed after Rnd3 knockdown in A549 or H460 cells. Taken together, our data indicate that knocking down Rnd3 in A549 and H460 cells does not increase ROCK1 signaling. Interestingly, despite not increasing ROCK1 signaling or actin stress fiber formation, Rnd3-depleted lung adenocarcinoma cells still have significantly decreased cell migration and invasion rates, indicating an alternative pathway controlling these processes.

To further support this claim, we tested whether ROCK1 inhibition could rescue the decreased metastatic potential of Rnd3-depleted lung adenocarcinoma cells. Klein et al show that inhibiting ROCK1 with Y-27632 (Ishizaki et al, 2000) rescues the decreased migration and invasion rates observed after Rnd3 knockdown in melanoma cells, indicating that Rnd3 functions through ROCK1 (Klein & Aplin, 2009). This makes ROCK1 a potential therapeutic target to control cell migration and invasion (Barcelo et al, 2023). Here, we were unable to rescue the Rnd3-knockdown phenotype of decreased cell invasion or migration in A549 or

H460 cells with ROCK1/2 inhibition. We did observe the characteristic robust loss of actin stress fibers after 5 μM Y-27632 treatment in both control and Rnd3-knockdown cells, and an increase in the invasion rates of control H460 cells, indicating successful inhibition of ROCK1 with Y-27632. However, there was no significant effect on cell migration or invasion of Rnd3-knockdown cells treated with Y-27632. Furthermore, increasing Y-27632 concentration to 10 μM had no additional effect on actin stress fiber number or invasion rates of A549 or H460 cells. Y-27632 is commonly used as a ROCK1 inhibitor, but it also inhibits ROCK2 and PRK2 activity even at the recommended concentration of 5 μM; however, this lack of specificity and off-target effect also increases with increased concentration (Davies et al, 2000; Ishizaki et al, 2000). ROCK1 and ROCK2 share 64% homology but have distinct and sometimes opposing roles (Lock et al, 2012; Julian & Olson, 2014). To determine whether Rnd3 was acting via the ROCK1 pathway in A549 and H460 cells and that we were not experiencing off-target effects by inhibiting both ROCK1 and ROCK2 (and PRK2) with Y-27632, we specifically knocked down ROCK1 and ROCK2 separately and in combination with Rnd3. We first tested this system in melanoma cells, in which Klein et al reported that ROCK1/2 inhibition with Y-27632 rescued the Rnd3-knockdown phenotype of decreased migration, restoring migration rates to WT (Klein & Aplin, 2009). Our data corroborate Klein et al, and we showed that knocking down either ROCK1 or ROCK2 in Rnd3-knocked down melanoma cells rescued the invasion rate back to WT. We then performed the same experiment in the lung adenocarcinoma cells (A549 and H460); however, neither ROCK1 nor ROCK2 knockdown in Rnd3-knockdown cells fully rescued invasion or migration rates in either A549 or H460 cells. We also knocked down ROCK1 individually and in combination with Rnd3 in the PDLBM cells and again did not observe a rescue of the invasion phenotype. In addition, knocking down either ROCK1 or ROCK2 in Rnd3-knockdown cells had no significant effect on their invasion or migration rates when compared to Rnd3-knocked down cells alone, for both the lung adenocarcinoma cell lines and PDLBM cell lines. Taken together, these data indicate that Rnd3 depletion in lung adenocarcinoma cells does not result in increased ROCK1 activity, and the Rnd3-dependent regulation of cell invasion and migration occurs via a ROCK1/2-independent pathway in both lung adenocarcinoma and patient-derived lung-to-brain metastasis cell lines. This highlights the importance of understanding and defining this pathway to identify potential therapeutic targets, which regulate the metastatic potential of lung adenocarcinoma cells.

RhoA is a known mediator of the actin cytoskeleton and cell migration and commonly signals via ROCK1 to regulate these processes. However, RhoA can also signal via alternative pathways to modulate the actin cytoskeleton and cell migration (Ridley, 2013; Schaks et al, 2019). This is also reflected in our data as we observed that inhibiting or knocking down ROCK1, a downstream target of RhoA, did not affect cell migration or invasion in most of the cell lines tested in this study. Rnd3 can inhibit RhoA activity by binding to and recruiting p190RhoGAP to RhoA, resulting in RhoA inactivation (Wennerberg et al, 2003); therefore, in Rnd3-knockdown cells there is the potential for increased RhoA activity, which may affect migration and invasion rates. Knocking down RhoA resulted

in a significant decrease in cell migration and invasion compared with control cells, indicating additional signaling mechanisms outside of ROCK1 signaling. However, when we performed the Rnd3:RhoA double knockdowns, cell migration in A549 cells or the invasion rate in H460 cells was not rescued back to WT rates or reduced further beyond individual knockdowns. These data suggest either that Rnd3 is not signaling via RhoA to regulate cell migration and invasion, or that loss of RhoA signaling has an alternative dominant phenotype on these biological processes.

Cell migration and invasion can also be influenced by cell adhesion to the extracellular matrix (Trepat et al, 2012). Integrins are adhesion molecules that provide mechanical attachment between the cell and the extracellular matrix and mediate signal transduction to control proliferation, differentiation, and cell migration (Aksorn & Chanvorachote, 2019). There are 18 alpha and 8 beta subunits, making 24 distinct heterodimer combinations. Integrins are transmembrane proteins that facilitate cell adhesion by forming adhesion complexes (focal adhesions and hemidesmosomes), which link the cells' actin or intermediate filament cytoskeletons to the extracellular matrix (Baster et al, 2025). Altered integrin expression, activation, and/or heterodimer composition have been implicated in regulating cell adhesion and signal transduction to control cell migration and invasion (Aksorn & Chanvorachote, 2019; Baster et al, 2025). Rnd3 has been shown to regulate beta 1 integrin activation, with decreased Rnd3 expression resulting in increased beta 1 integrin activation leading to increased cell adhesion (Liebig et al, 2009; Endzhievskaya et al, 2023) and decreased directional cell migration (Endzhievskaya et al, 2023). Therefore, we postulated that loss of Rnd3 expression may affect integrin regulation, which in turn could potentially alter cell adhesion or signal transduction pathways, culminating in our observed decreased cell migration and invasion phenotype. We investigated the expression levels of a small panel of integrins in our Rnd3-knocked down A549 cells, to include integrins which were both expressed in lung adenocarcinoma cells and have also been implicated in regulating cell migration. We were specifically interested in beta 1, as previous work by Endzhievskaya et al (2023) has shown a link to Rnd3 and beta 1 activation (Endzhievskaya et al, 2023). Included in our panel was beta 1's common heterodimer partner, integrin alpha 5, as alpha5:beta1 heterodimers have been implicated in regulating cell migration and invasion (Aksorn & Chanvorachote, 2019). In line with published data, we did not observe a change in beta 1 expression after Rnd3 knockdown (Liebig et al, 2009; Endzhievskaya et al, 2023); however, we observed a significant increase in alpha 5 expression levels. From our limited panel, we observed an increase in alpha 5 and a decrease in beta 3 expression; for this study, we opted to focus on increased alpha 5 expression. However, it is important to note that our data suggest loss of Rnd3 is affecting the expression of other integrins, so future studies will be required to expand the panel of integrins to investigate their involvement in Rnd3's regulation of cell migration and invasion in lung adenocarcinoma cells. Knocking down WT alpha 5 expression levels in A549 cells alone had no effect on cell migration or invasion rates. However, knocking down alpha 5 expression levels in the Rnd3-depleted A549 cells resulted in the restoration of cell migration and invasion rates, indicating that the

increase in alpha 5 expression retards cell migration and invasion. These data indicate that Rnd3 is modulating alpha 5 expression to regulate cell migration and invasion in lung adenocarcinoma cells. How loss of Rnd3 expression results in increased alpha 5 expression and how this increase in alpha 5 levels is affecting cell migration and invasion are currently unknown. We postulate that the loss of Rnd3 and subsequent increase in alpha 5 protein levels may alter the cells' ability to adhere to the extracellular matrix. How Rnd3 is modulating alpha 5 and beta 3 expression is unknown, and loss of Rnd3 could be affecting signaling pathways regulating transcription or stability of these proteins. For future studies, we will expand our integrin panel, and perform functional studies to assess cell adhesion, as well as colocalization studies of alpha 5 integrins in the Rnd3-knockdown cells. These data will aid in defining the mechanism behind the Rnd3-alpha 5 regulation of cell migration and invasion in lung adenocarcinoma cells.

In summary, we showed that patients with lung adenocarcinoma who express low levels of Rnd3 had significantly higher survival probability rates; a hypothesis for this is that a lack of Rnd3 expression reduces the metastatic potential in these cells. We identified that knocking down Rnd3 expression in both lung adenocarcinoma and patient-derived lung-to-brain metastasis cells significantly decreases cell migration and invasion, two hallmarks of metastasis. We also identified that in lung adenocarcinoma cells, Rnd3 regulates cell migration and invasion not by RhoA-ROCK1 signaling but in part by a novel mechanism modulating levels of the cell adhesion molecule, alpha 5 integrin. These data open up the field to identifying alternative mechanisms and pathways for how Rnd3 is regulating cell invasion and migration in lung adenocarcinoma cells, as well as having important implications on the use of integrin blocking antibodies to control metastasis.

# Materials and Methods

### TCGA analysis

Transcript data from TCGA-LUAD (The Cancer Genome Atlas Lung Adenocarcinoma) dbGaP study accession number phs000178 were downloaded using the "TCGAbiolinks" package in R. N = 501 patients after selecting for those with Rnd3 expression and survival data. Raw count data were processed, normalized, and converted to log scale in R using the "DESeq2" package. The "survminer" R package was used to determine the optimal cut point for Rnd3 expression; the package uses the maximally selected rank statistics from the "maxstat" R package. Approximately 34% (N = 172/501) of patients had high expression of Rnd3, and 66% (N = 329/501) had low expression of Rnd3. Top and bottom quartile cutoff groupings of Rnd3 expression, comparing survival probability rates of lung adenocarcinoma patients with high Rnd3 top 25% (n = 126/501) versus low Rnd3 bottom 25% (n = 126/501) expression, were also performed. Kaplan–Meier plots were generated and visualized using the "survival" and "survminer" R packages to compare the overall survival of patients with high Rnd3 versus low Rnd3 groupings.

### Cell culture

Authenticated A549 (CCL185) and H460 (HTB-177) lung adenocarcinoma cells and A375 (CRL1619) melanoma cells were obtained from the ATCC. For primary patient-derived brain metastatic non–small-cell lung carcinoma cells, PDLBM (CI34601 [601] and CI35620 [620]), informed consent was obtained from all subjects and experiments conformed to the principles set out in the WMA Declaration of Helsinki and the Department of Health and Human Services Belmont Report and approved by University of Arkansas for Medical Science's Institutional Review Board (#228443; IRB) accredited by the Association for the Accreditation of Human Research Protection Programs (AAHRPP). PDLBM cell lines (CI34601 [601] and CI35620 [620]) were generated from tumors obtained within a few hours of surgical removal and bio-banked at UAMS Tissue Biorepository and Procurement Core. CI34601 was derived from a 56yo African American male. CI35620 was derived from a 52yo Caucasian male. Each specimen was determined to be consistent with brain metastasis from lung primary by a board-certified neuropathologist. A cell line CI34601 has been characterized by the cancer dependency map (DepMap ID: ACH-002996). A549, H460, 601, and 620 cells were cultured in RPMI 1640 medium with GlutaMAX supplement (Thermo Fisher Scientific), 601 and 620 were cultured with the addition of 1% glucose (Thermo Fisher Scientific), and A375 cells were cultured in DMEM high-glucose medium (Thermo Fisher Scientific), all containing 10% heat-inactivated fetal bovine serum (Hyclone Laboratories). Cells were cultured in a humidified incubator at 37°C with 5% $CO_2$. For treatments with the ROCK1/2 inhibitor Y-27632 (Cell Signaling Technology), cells were treated with 5 or 10 $\mu M$ Y-27632 or equivalent concentration of DMSO as a control, and the medium was replaced with fresh medium containing Y-27632 or DMSO every 16 h during assays.

### Transient transfection with siRNA oligonucleotides

Cells were transiently transfected with siRNA oligonucleotides using Lipofectamine RNAiMAX (13778; Invitrogen) as described in the manufacturer's instructions. siRNA oligonucleotides were purchased from Horizon Discovery: Rnd3(A) (J-007794-07) UAGUAG AGCUCUCCAAUCA, Rnd3(D) (J-007794-06) CUACAGUGUUUGAGAAUU A, ROCK1 (J-003536-06) CUACAAGUGUUGCUAGUUU, ROCK2 (J-004610-06) GCAACUGGCUCGUUCAAUU, RhoA(1) (J-003860-13) GGA AUGAUGAGCACACAAG, RhoA(2) (J-003860-12) GCAGAGAUAUGGCAA ACAG, Alpha 5(A) (J-008003-08) GAACGAGUCAGAAUUUCGA, and Alpha 5(B) (J-008003-09) UCACAUCGCUCUCAACUUC. NSC oligonucleotides were purchased from QIAGEN (SI03650325). Cells were transfected for 24–48 h before assays. Cells were lysed at the end of each assay for Western blot analysis.

### Western blot analysis

Cells were lysed in RIPA buffer (Thermo Fisher Scientific) containing HALT phosphatase and protease inhibitors (Thermo Fisher Scientific) and sonicated. Protein (15 $\mu g$) was loaded onto a 4–12% gel and separated by SDS–PAGE, then transferred onto an IBlot2 PVDF membrane (Invitrogen). Primary antibodies used in

this study were as follows: Rnd3 (3664), ROCK1 (4035), RhoA (2117) pMLC2 (3671), pCofilin (3313), Cofilin (5175), alpha 5 (4705), alpha V (4711), beta 1 (9699), beta 3 (13166), beta 5 (3629) integrins, and tubulin (2146) (Cell Signaling Technology); ROCK2 (610623; BD Biosciences); and MLC2 (MABT180MI; Millipore). Secondary antibodies used in this study were as follows: horseradish peroxidase–conjugated anti-rabbit (7074) and anti-mouse (7076) (Cell Signaling Technology). Protein was detected with the ECL substrate (Thermo Fisher Scientific) and imaged with a ChemiDoc (Bio-Rad) imaging system with Image Lab software (Bio-Rad).

## Densitometry of Western blots

Densitometric analysis was carried out using ImageLab software (Bio-Rad) and was performed on multiple Western blots representing samples used in specific assays to determine whether significant changes in expression occurred following treatments. For each set of Western blots (protein of interest and the corresponding tubulin blot), protein lanes and bands of interest were defined with band area remaining constant. Adjusted total volume band intensity (adjusted for local background), for each isolated band, was determined by ImageLab software. For each sample, the expression level of the protein of interest was normalized to its own tubulin loading control, then normalized to the expression level of the protein of interest in the control sample. Data are presented as individual mean for each experiment with bar representing overall mean ± SEM. Statistical comparisons were performed using one-way ANOVA with Dunnett's multiple test correction. At least three biological repeats were carried out for each densitometric analysis (see the figure legend for actual n number for each experiment).

## Microscopy

Cells were seeded onto glass coverslips 24–48 h before fixation with 4% PFA in PBS solution for 10 min at room temperature; cells were permeabilized in 0.2% Triton X-100/PBS solution for 5 min at room temperature and stained with Hoechst 33342 and phalloidin (4082 and 12877; Cell Signaling Technology) for 30 min in the dark at room temperature. For alpha 5 staining, cells were incubated with alpha 5 integrin antibody (150361; Abcam) for 1 h, then incubated with anti-rabbit-488 (111-545-144; Jackson Labs) plus Hoechst 33342 and phalloidin for 1 h. Coverslips were mounted onto glass slides using mounting medium (Dako). For epifluorescence, cells were visualized with a Nikon Ts2-FL inverted microscope equipped with a Panda sCMOS camera and images were captured with Nikon NIS-Elements.

Actin stress fiber number quantification was measured from five fields of view (FOV) imaged using a Nikon Plan Apo Lambda D 60x oil objective; the total number of actin stress fibers per cell in five cells per FOV was counted for each condition, over at least three biological repeats. Average actin stress fiber number per cell was calculated and normalized to the number of actin stress fibers per cell in the NSC DMSO-treated condition. Data were plotted as individual mean for each experiment and normalized to own DMSO-treated NSC, with bar representing overall mean ± SEM.

Alpha 5 integrin fluorescence intensity was measured using the regions of interest (ROIs) tool in Nikon NIS-Elements. Each image was captured using a Nikon S Plan Fluor ELWD 40x air objective, using the same exposure time to capture alpha 5 staining in each condition and biological repeat. Five FOVs were captured per condition, 10 individual cells plus a section of background per FOV were defined as ROIs in Nikon NIS-Element software, and the average intensity for each cell minus background for each FOV was determined (>50 cells/condition/biological repeat). The mean intensity was calculated for each condition per biological repeat, data were presented as individual cells with bar representing overall mean for each biological repeat ± SEM from n = 4 assays, and one-way ANOVA with Dunnett's multiple test correction compared with NSC was performed. Representative images were captured using a Nikon Plan Apo Lambda D 60x oil objective using the same exposure time to capture alpha 5 staining in each condition.

## Cell viability analysis

Cells were transiently transfected (see above) for 24 h before being reseeded in triplicate into a 96-well plate at $2 \times 10^3$ cells and incubated for 24, 48, or 96 h before the addition of Cell Counting Kit-8 reagent (Dojindo), incubation (4 h), and plate reading were done as described in the manufacturer's instructions. The mean percentage, ± SEM, of viable cells was analyzed from n = 4 assays and normalized to NSC cells. One-way ANOVA with Dunnett's multiple test correction compared with NSC was performed.

## Cell cycle analysis

Cells were transiently transfected (see above) for 48 h before cell cycle analysis. Cells were washed with PBS, fixed with 70% ethanol for 1 h at −20°C, washed with PBS, treated with RNase A (final concentration 0.5 mg/ml) for 1 h at 37°C, stained with propidium iodide (final concentration 10 μg/ml), and incubated at 37°C for 1 h. Cells were subjected to flow cytometry with a FACSVerse (Becton Dickinson). Data were analyzed with FCS Express software (De Novo). The mean percentage, ± SEM, of cells in each phase of the cell cycle was analyzed from n = 4 assays. For G1 and G2/M comparison, one-way ANOVA with Dunnett's multiple test correction compared with NSC was performed.

## Cell migration assay

Cells were seeded into a p6 well and reached 100% confluence over the course of 48 h. At 48 h, the monolayer of cells was scratched with the tip of a 1-ml pipette tip and imaged at 0, 4, 8, 24, and 48 h. For each "wound," five images were captured with Nikon NIS-Elements software at each time-point using a Nikon Ts2-FL inverted microscope equipped with a Panda sCMOS camera and Nikon Plan Fluor 10x air objective. Images were analyzed with ImageJ/Fiji software. A grid was applied to each image as a reference guide, and the measure tool was used to manually measure the wound width from one wound edge to the other wound edge; five measurements were taken per image along this equally space grid. For each time-point, the mean wound width was calculated

and percentage wound closure at each time-point was compared with mean wound width immediately after wound formation (t = 0 h). At the endpoint, lysates were collected to confirm knockdown using Western blotting. End time-points of 24 h for A549 and 48 h for H460 cells were chosen because of natural differing basal rates of migration. After our optimization experiments, A549 and H460 control cells reach ~60% wound closure at 24 and 48 h, respectively. These conditions were then maintained throughout, as this ~60% closure of control cells allows us to observe either an increase or a decrease in migration rate.

### Transwell invasion assay

Corning Matrigel matrix (CB-40234; Thermo Fisher Scientific) was used to mimic the extracellular matrix to assay the invasion capacity of cells. 12-well plate transwell inserts with 8-$\mu$M pores (877112; Thermo Fisher Scientific) were coated with 200 $\mu$l 300 $\mu$g/ml Matrigel diluted in ice-cold coating buffer (0.01 M Tris, pH 8; 0.7% NaCl) and allowed to cure for 2 h at 5% $CO_2$ at 37°C. Knockdown cells were transiently transfected with siRNA oligonucleotides (see above) and incubated for 24 h before being seeded at 3.0 × $10^5$ cells/500 $\mu$l onto the Matrigel-coated transwells; an additional 1.5 ml complete growth medium was added to the well of the companion plate housing the transwell insert. Cells were allowed to invade for 24 or 48 h before fixation and staining with DipStain (Thermo Fisher Scientific). Cells still present inside the transwell were removed with a cotton swab. Invaded cells were imaged directly from transwells using a Leica DMi1 inverted microscope equipped with a Leica Hi Plan 10x air objective, and captured using Leica Application Suite software V4.12. Five FOVs per condition were taken at 10× magnification, all invaded cells were counted, the mean number of invaded cells per condition was calculated, and the percentage of invaded cells was generated by normalizing to the number of invaded NSC cells, setting the mean number of invaded NSC cells as 100% invasion. Data were plotted as individual mean for each biological repeat, and normalized to own NSC, with bar representing overall mean ± SEM.

End time-points of 24 h for A549 and 48 h for H460 cells were chosen because of natural differing basal rates of invasion. After our optimization experiments, we achieved 50% confluence of the invasion plate for control A549 cells at 24 h and H460 48 h. These conditions were then maintained throughout, as this ~50% confluency of control cells allows us to observe either an increase or a decrease in invasion rate.

### Statistical analysis

All statistical analyses were performed with PRISM (GraphPad) unless otherwise stated. Statistical significance was determined with one-way ANOVA with Dunnett's multiple test correction compared with NSC cells unless stated. Statistical significance was defined as $P < 0.05$.

## Data Availability

Raw data underlying this work are available from the corresponding author upon reasonable request. Patient expression and survival data (TCGA-LUAD) used in this study are publicly available from The Cancer Genome Atlas.

## Supplementary Information

## Acknowledgements

We would like to acknowledge the support provided during preparation of this article by the Center for Childhood Obesity Prevention's Scholarly Writing Program funded by the National Institute of General Medical Sciences of the National Institutes of Health under Award Number P20GM109096 (Arkansas Children's Research Institute, PI: Weber). This project was supported by the Winthrop P Rockefeller Cancer Institute at the University of Arkansas for Medical Sciences (280-3013042) (KR Ryan); by a Winthrop P Rockefeller Cancer Institute Seeds of Science Grant (AWD00055290) from the Little Rock Envoys (KR Ryan), and Barton Pilot Awards (AWD00055285 and AWD00056089) from the College of Medicine (KR Ryan), all from University of Arkansas for Medical Science; and by the National Institute of General Medical Sciences of the National Institutes of Health under Award Number P20 GM152281. This research used resources from the Center for Molecular Interactions in Cancer (CMIC) Structural Biology Core, as well as the CMIC Biomolecular Interactions Core, funded by P20 GM152281 from NIGMS.

### Author Contributions

N Garcia Garcia: data curation, validation, investigation, methodology, and writing—review and editing.
THV Nguyen: conceptualization, validation, investigation, methodology, and writing—review and editing.
D Richey: investigation and writing—review and editing.
EF Cox: investigation and writing—review and editing.
J Coca Juaristi: validation, investigation, and writing—review and editing.
C Ashby: formal analysis, methodology, and writing—review and editing.
A Rodriguez: conceptualization, resources, validation, and writing—review and editing.
KR Ryan: conceptualization, data curation, supervision, funding acquisition, validation, investigation, visualization, methodology, and writing—original draft, review, and editing.

### Conflict of Interest Statement

The authors declare that they have no conflict of interest.

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
