## [Reviewer comments · Life Science Alliance]

Rnd3 regulates lung cancer cell invasion and migration independently of ROCK1 via alpha 5 integrin

Noemi Garcia Garcia, Thanh Nguyen, Dane Richey, Emily Cox, Jon Coca Juaritsti, Cody Ashby, Analiz Rodriguez, and Katie Ryan

DOI: <https://doi.org/10.26508/lsa.202503494>

Corresponding author(s): Katie Ryan, University of Arkansas for Medical Sciences

Review Timeline:

Submission Date:	2025-08-21
Editorial Decision:	2025-10-07
Revision Received:	2026-01-03
Editorial Decision:	2026-02-02
Revision Received:	2026-02-06
Accepted:	2026-02-11

Scientific Editor: Sarita Hebbar

Transaction Report:

October 7, 2025

Re: Life Science Alliance manuscript #LSA-2025-03494-T

Dr. Katie Rose Ryan
University of Arkansas for Medical Sciences
Department of Biochemistry and Molecular Biology
University of Arkansas for Medical Sciences
Little Rock, Arkansas 72205-7101

Dear Dr. Ryan,

Thank you for submitting your manuscript entitled "Rnd3 regulates lung cancer cell invasion and migration independently of ROCK1 via alpha 5 integrin" to Life Science Alliance. The manuscript was evaluated by three experts whose comments are appended below. As you will note, all the three reviewers agree that your study is interesting and extends previous knowledge in this area. But they have raised significant concerns related to technical clarity and the need for corroboration of a few of the findings.

We agree the following points raised by the reviewers must be addressed before publication at LSA:

- All the three reviewers highlighted the low quality of images in particular in Figure 4 and 8. We agree that you must include clearer images and high-resolution images for both types of stainings. Further we also concur with Reviewer3 that you must validate the alpha5 integrin staining with alpha5 integrin knockdown cells.
- The reviewers have acknowledged your attempt in this work to link clinical data to experiments in cell culture. Reviewer 2 in point 1 makes relevant suggestions to clarify the clinical data explaining the correlation between Rnd3 levels and survival probability in LUAD. We agree that this must be addressed to strengthen this aspect of the study.
- Related to the description of stress fibres, Reviewer 1 has asked for quantification of stress fibres whereas Reviewer 2 has asked for cell shape quantification as a proxy since LUAD cells have few stress fibres. We leave it to you to include any quantification, either for stress fibres or for cell shape, but you must include both points in discussing the results.
- Reviewers 1 and 2 have asked for confirmation that the ROCK inhibitor Y27632 is active given the unexpected results in the context of cell invasion. Reviewer 2 has suggested a titration curve for the inhibitor in cell invasion assays (independent of Rnd3 levels) to show that the inhibitor is active at the concentration used. We agree that this must be included in the revised manuscript.
- The reviewers are consistent in their suggestion to include independent replicates for all western blotting data with associated quantification/scatterplots and statistical significance information. We concur with the reviewers that this inclusion is required for the revised manuscript.

In line with their overall assessment, we invite you to submit a revised manuscript. When submitting the revision, please include a letter addressing the reviewers' comments point by point. While a rebuttal must respond to all points in some form, additional experiments to resolve these points, other than indicated above, will not be required.

We offer to connect with you should you have any questions on our invitation to resubmit. Please respond if you would like to meet with suggested dates/time. We will be happy to send you a meeting link and talk to you.

Thank you for this interesting contribution to Life Science Alliance. We are looking forward to receiving your revised manuscript.

Sincerely,

Sarita Hebbar, PhD
Scientific Editor
Life Science Alliance

B. MANUSCRIPT ORGANIZATION AND FORMATTING:

Reviewer #1 (Comments to the Authors (Required)):

The study titled "Rnd3 regulates lung cancer cell invasion and migration independently of ROCK1 via alpha 5 integrin", reports several interesting observations. Authors report that low Rnd3 expression correlates with improved survival in patients and that Rnd3 knockdown reduces migration and invasion in LUAD cell lines and patient-derived brain metastasis cells via a mechanism that appears ROCK-independent and involves upregulation of $\alpha 5$ integrin in cells.

The major strength of this paper is the use of multiple cell lines, multiple siRNAs and rigorous attempt to link the data with clinically derived samples.

Minor remarks-

1. There are several grammatical errors throughout the paper, which need correction. Some of these are-
E.g.- a. We observed a significant decrease in cell invasion and cell migration, two hallmarks of metastasis, but no effect cell proliferation or cell death.
b. These data indicate that depletion of Rnd3 does not affect cell viability or proliferation has no significant change in cell number was observed over time following loss of Rnd3 expression.
c. This is also reflexed in our data has we observed that inhibiting or knocking down ROCK1
2. Its isn't clear what the rationale is behind letting A549s invasion assays being done for 24 hours, whereas H460s are done for 48 hours? Kindly explain.
3. Adding molecular weight marking for all protein western blots is mandatory and there is no excuse for not included this in the blot images.
4. Please also add scale bars to microscopy images.
5. Was DMSO used as the control for Y27632 treatment? Please clarify in the experimental protocol and when discussion the data.

Major remarks-

1. Knockdown studies are integral to the interpretations made and lack of quantitative evaluation of knockdowns with biological replicates is needed. Will be useful to share KD for all the sets of experiments used for migration and other functional studies. Quantitative scatter plots for WBs should be added.
2. Reconstitution with siRNA resistant mutants to evaluate major functional outcomes affected by knockdo3n are restored will be

useful to show.

3. Please add images for experiments done to evaluate cell migration by performing wound healing assays.
4. Does Y27632 inhibitor treatment affect the percentage invasion of cells in Rnd3 knockdown cells?
5. In evaluating the Rnd3 mediated regulation of ROCK1/2, autophosphorylation of ROCK1 at Ser1333 and of ROCK2 at Ser1366 could be evaluated to check their activation status in cells.
6. Image quality and resolution to show actin stress fibres in Fig. 4 B, C is not very clear. Also, in image 8 B, the image quality is poor. The localization of alpha 5 is not seen at the expected places. Will be useful to show insets, or better-quality zoomed in images.
7. Quantitation of the actin stress fibres is needed to make any comment on the number of stress fibres in Y27632 inhibitor untreated/ treated cells.
8. "Worm-like" actin structure is not a standard terminology. It will help to discuss these structures in detail, and highlight also cross- reference with literature.
9. Rnd3 knockdown causes a decrease in integrin beta 3 and increase in integrin alpha 5 integrin. While the authors have further established the role of integrin alpha 5 in regulating Rnd3 mediated migration and invasion, they have not commented on the possible role of integrin beta 3. Its possible implications are useful to comment on
10. The discussion section is a bit lengthy and can be made crisper and to the point. The authors could consider discussing the possible mechanistic involvement and role for $\alpha 5$ regulation (surface vs total, transcription vs trafficking) in regulating Rnd3 mediated migration and invasion in LUAD cell lines.

Reviewer #2 (Comments to the Authors (Required)):

This is an interesting manuscript which builds on previous publications showing the contribution of Rnd3 to cancer migration and invasion and that higher Rnd3 expression correlates with worse prognosis in several cancer types, including lung cancer as further reported here. The authors find that lung cancer-derived cell lines depleted of Rnd3 have higher $\alpha 5$ integrin expression, and that decreased $\alpha 5$ integrin expression reverses the effects of Rnd3 depletion on migration and invasion using standard in vitro models. This extends on a previous study reporting that Rnd3 regulates integrin expression. The manuscript does not explain the molecular basis for this integrin upregulation, which could be investigated in the future.

General interpretation. Rnd3 is known to be phosphorylated by ROCK1, which inhibits its activity, but whether it signals through inhibiting Rho/ROCK signaling is not consistently established across different models, nor is the molecular basis for any effect fully understood. The authors should consider their wording on this topic carefully, giving both sides of the argument.

1. Figure 1. Explain the cut off for Rnd3 high/low expression in patients -what cut off was chosen? This appears to have been chosen to be optimal for observing an effect. What happens if you use different cut-offs or just use the patients with the highest expression and lowest expression (e.g. top 25% and bottom 25%)? Add a graph showing the range of Rnd3 expression in the patients and where the cut-off is. This appears to be replicating previous data - was that using a different set of patient samples? More information is needed in the text and a new graph.
2. It is not surprising that Rnd3 depletion decreases migration and invasion of the cell lines used in this study. This has been demonstrated in multiple studies over many years in a wide variety of cell lines (or that increased expression increases invasion). The type of cell lines used is not important because they are all cultured in vitro on hard plastic surfaces. All that differs for the two brain metastasis lines is possibly that they have been cultured for less time in vitro than the two long-standing lung adenocarcinoma cell lines used. The authors should state specifically that these data are consistent with many other studies.
3. The effect of Rnd3 depletion on cell proliferation/cell cycle progression depends on the cell line. It is not really surprising that highly proliferative cancer cells are not affected by Rnd3 depletion in these assays, because they are already likely to have mutations that bypass regulation of cell cycle progression. Please comment on whether these lines have such mutations e.g. by checking DEPMAP portal of cancer cell lines.
4. Figure 2A: provide a graph quantifying Rnd3 protein levels over at least 3 independent experiments.
5. For all graphs, provide absolute p values so that readers can judge for themselves the level of significance. This is considered best practice now.
6. Figure 4 and Figure S1. To prove that Y27632 is active in the author's assays, biochemical evidence is required. Cofilin phosphorylation can be mediated by multiple kinases and is not a ROCK-specific target. pMLC changes are also indirect because ROCKs actually phosphorylate pMYPT, which should be tested as well to be sure that the ROCK inhibitor is working correctly (and whether Rnd3 knockdown alters ROCK1 signaling. Before concluding that it has no effect on invasion (which is surprising) the authors need to demonstrate the inhibitor is active at the concentration they use. They should also titrate Y27632 to check whether higher concentrations affect invasion (alone, irrespective of +/- Rnd3).
7. The single images shown in Figure 4 +/- Y27632 and Rnd3 depletion are very difficult to interpret, since the lung adenocarcinoma cell lines have few visible stress fibers (not surprising as most cancer cell lines have few stress fibers). Cell shape should be quantified instead to determine if Y27632 and/or Rnd3 depletion are having expected effects.
8. Figure 5. The authors should change the wording concerning the effects of knocking down ROCK1 and ROCK2 on the Rnd3 migration/invasion phenotypes, given that ROCK1/2 knockdown is not very strong yet they observe a significant effect in one of the two lung cancer cell lines tested. They cannot conclude that the Rnd3 knockdown phenotype is independent of ROCK1/2 given these data, but that it acts partially through ROCK1/2. See also point above that they have not conclusively proven that

Y27632 is active in their experiments.

9. Graphs in Figures 5 and 6 do not show the direct statistical comparison between Rnd3 depletion and joint Rnd3/ROCK1 or ROCK2 depletion, and in Figure 7 between Rnd3 depletion and Rnd3/RhoA. The same holds for supplemental figure graphs. These seem the most relevant comparisons. In any case, tables showing all the statistical comparisons (every condition against every other condition) should be provided with absolute p values. The authors will have these data from the ANOVA tests.

10. Figure 7: the authors should interpret these results with caution because RhoC can also activate ROCK1/ROCK2 and could be the most important Rho subfamily member in these cell lines. It is clear from several studies that RhoC depletion strongly reduces invasion and metastasis, whereas the effect of RhoA depletion is variable depending on the cell type. Moreover, since depletion of RhoA in these cells reduces cell migration, how would they expect it to rescue the effect of Rnd3 depletion on cell migration? It should be synergistic in decreasing migration, if anything.

11. Figure S2 requires quantification of western blots for each integrin over multiple experiments to demonstrate reproducibility of the results.

Minor points:

1. There are grammatical errors throughout the manuscript that need to be corrected.

2. There are inconsistencies in how 'hours' are abbreviated throughout. The accepted abbreviation is h (not hrs).

3. Scale bars are missing and need to be added to all images.

4. Figure 3, give absolute n values for each condition rather than >4.

5. Y27632 inhibits ROCK1, ROCK2 and PRKs (see papers testing multiple kinase inhibitors on multiple kinases from the Cohen laboratory in Biochemical Journal). This should be stated in the text and every time it is used, state that it is at least a ROCK1/2 inhibitor (not just ROCK1).

Reviewer #3 (Comments to the Authors (Required)):

In this work, Garcia-Garcia et al. show that Rnd3 high expression correlates with a worse prognosis in lung cancer patients. They showed that Rnd3 mediates cell migration and invasion in 2 different cell lines of lung adenocarcinoma without significant effects on proliferation or viability. Since Rnd3 inhibits ROCK1/2 signalling, they went on to test whether Rnd3 could be mediating cell migration and invasion via this pathway. Inhibiting or silencing ROCK1 had no effect on the invasion or migration, and thus it seems like Rnd3 is controlling cell migration and invasion through a different pathway. Finally, the authors show that there is an upregulation of alpha5-integrin upon Rnd3 silencing and this upregulation prevents the cells from migrating/invading. Since in the literature there are reports of Rnd3 being either a tumor suppressor or tumor promoter, this work becomes relevant and the use of patient-derived metastatic cells reinforces the main point. Overall, the study is conducted properly and data supports the main conclusion; however, some aspects of the mechanism, discussion and the results presentation need to be improved.

1) The data strongly supports a role for Rnd3 in cell migration and invasion in lung adenocarcinoma.

2) The data strongly shows that Rnd3 mediates LUAD migration and invasion through a ROCK-independent pathway.

3) The data suggests that Rnd3 might repress expression of integrin alpha5, and that alpha5-integrin expression prevents LUAD cell migration/invasion. This is surprising since alpha5-integrin is known for mediating cancer cell invasion in other cancers. Moreover, the staining shown in the results is difficult to see and looks very unusual. The authors need to replace the IF picture of alpha5-integrin staining for a higher resolution picture to show localization or absence of alpha5-integrins at adhesion sites and/or cell-cell contact sites, and validate the specificity of the staining against alpha5-knockdown cells. Moreover, they should indicate their rationale for looking at the integrins they are showing, and include quantification of the blots and data showing expression of alpha6 and alpha2 to understand if the effect is specific to alpha5.

4) Specific points to address regarding the results, text and figures:

-In the introduction, 3rd paragraph, page 2, the authors mention that "ROCK1, in turn, can phosphorylate Rnd3, increasing Rnd3 levels and producing a negative feedback loop(Riou et al., 2013)." Here, the authors should also reference Komander 2008, EMBO J, doi: 10.1038/emboj.2008.226

-Please correct typos in page 3: "Rnd3 has been proposed to regulate many different cellular processes that are commonly dysregulated in cancer" and "These data suggest that Rnd3 may regulated the metastatic potential..."

-Add "that" in the following sentence: "We utilized patient derived lung-to brain metastasis (PDLBM) cells as a proxy for advanced disseminated disease and observed that knocking down Rnd3 expression in these cell lines resulted in a significant decrease in cell invasion"

-In page 4, this sentence should be deleted since the overall conclusion for the section is stated in the next paragraph: "These

data indicate that depletion of Rnd3 does not affect cell viability or proliferation has no significant change in cell number was observed over time following loss of Rnd3 expression."

-In the discussion, please mention that the studies showing a role for Rnd3 in proliferation have been done in fibroblasts or other cell types, and not in LUAD cells to help the reader put the previous work in context.

-References: DOI accession number for Wennerberg, K., et al. (2003), is incomplete. Please add the correct DOI link.

-The authors mention in the manuscript that they only looked at a small panel of integrins. What is the rationale for choosing those integrins? What is the composition of the extracellular matrix in the lungs? Which integrins would you expect to have in normal tissue and do those relate to the ones you looked at? If expressed in lung cells, the authors need to check alpha2-integrin and alpha6-integrin since those are the major collagen and laminin receptors.

-Please check and rephrase the second paragraph of the discussion (page 8). It contains some grammatical errors that make it difficult to read.

For example, the first sentence stating: "To investigate the potential mechanistic basis by which patients with low Rnd3 expression experience improved survival outcomes" makes the reader think that you would follow up patient outcomes and relate that data with tumor-derived organoids in real time, but this is not the case. A more suitable way to say this would be: "To investigate whether Rnd3 confers metastatic properties to lung cancer cells..." or ... "whether Rnd3 controls metastasis and cell invasion".

Also, at the end of the paragraph, please replace the word "suspectable" for "susceptible" if that is what the authors intended to say.

-In the discussion (page 9, 2nd paragraph), the authors state that "Surprisingly, no increase in ROCK1 activity was observed..." However, they did not measure ROCK1 activity, but rather ROCK1 signalling. Please change the wording throughout this paragraph, and also in Figure S1.

-Moreover, the first part of this paragraph (page 9, 2nd paragraph) sounds repetitive (all these facts were already mentioned in the introduction). Rather than listing these facts, the authors should compare their results to what is in the literature: previous studies have been done in melanoma or fibroblasts, and here they report a different mechanism for lung adenocarcinoma. What does this mean in terms of cell migration and attachment to the extracellular matrix of the invading tissue? Are cells less/more contractile to colonize a softer tissue when they have high/low Rnd3? How is the brain ECM different to lung ECM? Is there anything known about Rnd3 blocking fibronectin deposition? What would be interesting to look at next? If the mechanism is ROCK independent, could the authors speculate on what other mechanisms could be mediating the invasive phenotype?

-For all figures, the authors should state if the western blots showed are representative of n-number of repeats. Moreover, whenever possible, the blots showed to validate silencing should be shown after the assay results.

-For Figure 3 and related text in the results, the authors should mention why did they use different timepoints for different cell lines? For panel C and D, please indicate that this number is per Field of View (FOV).

-Figure 5: the title of this figure is very long, and while it is very nice to show that you have reproduced the published results in melanoma cells (which also demonstrates the effect of the ROCK inhibitor), this could be skipped in the Figure title and only state that Rnd3 controls cell migration and invasion in a ROCK-independent fashion in LUAD cells. In addition, could the authors please reorder the results, showing the LUAD results first, and then the results from the melanoma cells, and indicate in the legend that this was used as a control?

-Figure 8: The IF staining for alpha 5-integrin seems a bit strange, almost absent from focal adhesions and more localized to cell-cell contact sites. Is this correct? Could the authors please provide a higher resolution picture and validate their alpha 5-integrin staining against alpha5-integrin knockdown cells? Could they zoom into an area to show that alpha5-integrin is (or is not) at focal adhesion sites?

-Following up from the possible localization of alpha5 integrin at cell-cell junctions, do the Rnd3 knockdown cells cluster more than the NSC cells, similar to what has been observed in Ryan et al., 2012 (doi: <https://doi.org/10.1242/jcs.101931>)?

-Figures S1 and S2. Could the authors please show a quantification of the blots?

-For all figures, please clearly indicate that you are using siRNA. For example, as siRnd3(A), siROCK1, so that it is easier for the reader to understand the plots and figures.

Point-by-point combined and editor's comments:

We agree the following points raised by the reviewers must be addressed before publication at LSA:

-All the three reviewers highlighted the low quality of images in particular in Figure 4 and 8. We agree that you must include clearer images and high-resolution images for both types of stainings. Further we also concur with Reviewer3 that you must validate the alpha5 integrin staining with alpha5 integrin knockdown cells.

We apologize for the low-quality images. We have included larger, high-resolution images of actin staining in A549 and H460 control and Rnd3 knockdown cells treated with either DMSO control or the Y-27632 ROCK1/2 inhibitor in Figure 4A-B. We have also added additional actin staining images in supplemental Figure S2C-D. For images in Figure 8, integrin Alpha 5 staining, we have included large high-resolution images. For validation of the Alpha 5 staining, we have included images and quantification of Alpha 5 fluorescent intensity in control, siRnd3, siAlpha 5 and siAlpha5:Rnd3 double knockdown A549 cells. These data show both a significant increase in Alpha 5 staining in Rnd3 knock down cells (Figure 8A-B, S8D). Importantly, we also observed a significant decrease in Alpha 5 staining compared to NSC cells in conditions in which Alpha 5 has been knocked down, siAlpha5 and siRnd3:Alpha5 (Figure 8A-B), indicating the staining observed in NSC and Rnd3 knockdown cells is that of Alpha 5 staining. This information has been added to the main text in the manuscript.

-The reviewers have acknowledged your attempt in this work to link clinical data to experiments in cell culture. Reviewer 2 in point 1 makes relevant suggestions to clarify the clinical data explaining the correlation between Rnd3 levels and survival probability in LUAD. We agree that this must be addressed to strengthen this aspect of the study.

We apologize for the lack of information and clarity. The original cutoff groups for high and low Rnd3 expression were generated using the 'survminer' R package to determine the optimal cutpoint for Rnd3 expression; this package uses the maximally selected rank statistics from the 'maxstat' R package. This package finds the most statistically significant cutoff and is commonly used when identifying survival signatures as it utilizes all of the data set. We have now also included an analysis using the top and bottom quartiles and generated Kaplan–Meier plots for these groups, we observed patients with low levels of Rnd3 have significantly higher survival probability ($p = 0.0048$) (Figure S1A). We have also included the risk tables for each of the Kaplan–Meier plots (Figure 1, S1A), as well as the expression range plot denoting the cut-points for the groups utilized in the survival analysis (Figure S1B).

Although Li et al 2016 utilized the TCGA database their analysis is distinct from ours: They compared Rnd3 expression in normal lung tissue samples to squamous cell carcinoma and lung adenocarcinoma samples. Their high vs low Rnd3 survival analysis used a none specified sample set of 2437 lung cancer cases and 2435 controls, this may or may not contain the TCGA LUAD data set (comprising of 701 samples, down to 501 once matched normal samples have been removed) which we used for our analysis. However, we only utilized the TCGA LUAD data set for our analysis to investigate if Rnd3 expression levels correlated with patient outcome in lung adenocarcinoma specifically, our data is in line with other groups including Li et al which observe high Rnd3 expression correlates with poor patient outcome but have performed their analysis on different cohorts of lung cancer patient samples. We have made clear that our analysis is unique but is inline with other similar studies.

This additional information and data have been added to the manuscript.

-Related to the description of stress fibres, Reviewer 1 has asked for quantification of stress fibres whereas Reviewer 2 has asked for cell shape quantification as a proxy since LUAD cells have few stress fibres. We leave it to you to include any quantification, either for stress fibres or for cell shape, but you must include both points in discussing the results.

To address the reviewer's comments about actin stress fibers, we have added larger high-resolution images and have also quantified the number of stress fibers in both A549 and H460 control cells and Rnd3 knockdown cells with and without 5 μM treatment with the ROCK1/2 inhibitor, Y-27632. We observed a significant to near complete loss of actin stress fibers, following 5 μM of Y-27632 treatment in both control and Rnd3 knockdown cells, in both cell lines (Figure 4). To quantify actin stress fiber number per cell, we imaged 5 fields of view (FOV) using a 60x oil objective and counted the number of stress fibers in 5 cells per FOV for each condition, over 3 biological repeats, average stress fiber number per cell was calculated for each condition per biological repeat and was plotted mean \pm standard error of the mean (SEM) from $n = 3$ assays, one-way ANOVA with Dunnett's multiple test correction compared to NSC was performed.

In addition, we have also quantified the number of stress fibers in A549 and H460 cells treated with a titration of the ROCK1/2 inhibitor Y-27632. We compared 5 μM , the common working concentration of Y-27632, to 10 μM of Y-27632 with the equivalent volume of DMSO as a control. Increasing the concentration of the Y-27632 inhibitor pass 10 μM leads to additional off target effects through the inhibition of other kinases making data difficult to interpret. We argue that that 5 μM treatment with Y-27632 is inhibiting ROCK1 signaling in A549 and H460 cells as we observe a significant decrease in the number of stress fibers, to almost complete ablation, in both A549 and H460 cells and we observe no further decrease in the number of stress fibers following 10 μM treatment, indicating that stress fibers were abolished at 5 μM and doubling the concentration of Y-27632 had little addition effect on stress fiber number (**Figure S2**).

Although lung adenocarcinoma cell lines do have fewer stress fibers compared to fibroblast, we are able to observe and quantify both the presence and loss of stress fibers in these cell lines so we opted for this metric, we do also observed a change in cell shape following Y-27632 treatment but we have not quantified that. This information along with discussing cell shape, has been added to the main text in the manuscript and in Figure 4 and Figure S2.

-Reviewers 1 and 2 have asked for confirmation that the ROCK inhibitor Y27632 is active given the unexpected results in the context of cell invasion. Reviewer 2 has suggested a titration curve for the inhibitor in cell invasion assays (independent of Rnd3 levels) to show that the inhibitor is active at the concentration used. We agree that this must be included in the revised manuscript.

We believe the Y-26732 inhibitor is active and functioning to inhibit ROCK1. We observe a significant decrease in the number of actin stress fibers, to near complete abolishment of actin stress fiber, in both A549 and H460 NSC cells as well as Rnd3 knockdown cells following 5 μ M treatment of Y-27632. To support this we have included high-resolution images and quantification of actin stress fibers in Figure 4A-D. The loss of actin stress fibers is indicative of ROCK1 inhibition following treatment with Y-27632. Additionally, we also observe a significant increase ($p = 0.0103$) in cell invasion in NSC H460 cells treated with 5 μ M Y-27632 compared to DMSO-treated NSC H460 cells (**Figure 4F, H**). It is only the A549 cells in which 5 μ M Y-27632 had no significant effect on the invasion rates of NSC cells compared the DMSO-treated NSC A549 cells, there is a slight increase in both invasion and migration following Y-27632 treatment, but it is not significant (**Figure 4E, G, I-J**).

As requested, we performed a titration of the Y-27632 treatment in both A549 and H460 cells. We compared 5 μ M, the common working concentration of Y-27632, to 10 μ M of Y-27632 with the equivalent volume of DMSO as controls. Increasing the concentration of the Y-27632 inhibitor pass 10 μ M leads to additional off target effects through the inhibition of addition kinases making data difficult to interpret. First, we quantified the number of actin stress fibers in non-transfected A549 and H460 cells treated with either 5 μ M or 10 μ M of Y-27632 compared to their corresponding DMSO control treatments. Similar the data generated in the NSC A549 and H460 cells, we observed a significant decrease in the number of actin stress fibers, to almost complete ablation, in both non-transfected A549 and H460 cells following 5 μ M treatment of Y-27632, and observed no further decrease in the number of stress fibers following 10 μ M treatment, indicating that stress fibers were abolished at 5 μ M and doubling the concentration of Y-27632 had no additional effect on reducing stress fiber number (**Figure S2C-F**). We also performed invasion assays on A549 and H460 cells treated with either 5 μ M or 10 μ M of Y-27632, and their corresponding DMSO control treatments. We again observed no increase in invasion in A549 cells treated with either 5 μ M or 10 μ M of Y-27632, we did observe a significant increase H460 invasions rates following Y-27632 treatment however doubling the concentration of Y-27632 had no significant effect on the invasion rate compared to the standard 5 μ M treatment (**Figure S2G-H**). With the significant loss of stress fibers in both A549 and H460 cells, and the significant increase in cell invasion in H460 cells following 5 μ M of Y-27632 we conclude that Y-27632 is active and functioning to inhibit ROCK1 in these cell lines.

We have added the additional titration data (stress fiber quantification and invasion assay data) to the manuscript, Figure S2.

-The reviewers are consistent in their suggestion to include independent replicates for all western blotting data with associated quantification/scatterplots and statistical significance information. We concur with the reviewers that this inclusion is required for the revised manuscript.

We have performed and included densitometry on all Western blotting data associated with this manuscript, this includes independent biological replicates from samples used in function assays, validating either knockdowns or changes in protein expression. We have included all the corresponding plots and statistical significance in the main figures were possible or alternatively in supplemental figures. All previous claims of knockdown, decreased, increased, and no change in protein expression made in the original manuscript still stand and are now supported by the densitometric analysis. We have also included a description of how we performed the densitometry and analysis in the method section.

In line with their overall assessment, we invite you to submit a revised manuscript. When submitting the revision, please include a letter addressing the reviewers' comments point by point. While a rebuttal must respond to all points in some form, additional experiments to resolve these points, other than indicated above, will not be required. The typical timeframe for revisions is three months. Please note that papers are generally considered through only one revision cycle, so strong support from the referees on the revised version is needed for acceptance.

Below we have address each of the reviewers' individual comments:

Reviewer #1 (Comments to the Authors (Required)):

The study titled "Rnd3 regulates lung cancer cell invasion and migration independently of ROCK1 via alpha 5 integrin", reports several interesting observations. Authors report that low Rnd3 expression correlates with improved survival in patients and that Rnd3 knockdown reduces migration and invasion in LUAD cell lines and patient-derived brain metastasis cells via a mechanism that appears ROCK-independent and involves upregulation of $\alpha 5$ integrin in cells.

The major strength of this paper is the use of multiple cell lines, multiple siRNAs and rigorous attempt to link the data with clinically derived samples.

Minor remarks-

1. There are several grammatical errors throughout the paper, which need correction. Some of these are-
a. We observed a significant decrease in cell invasion and cell migration, two hallmarks of metastasis, but no effect cell proliferation or cell death.

Corrected to- "We observed a significant decrease in cell invasion and cell migration, two hallmarks of metastasis, but no effect on cell proliferation or cell death."

b. These data indicate that depletion of Rnd3 does not affect cell viability or proliferation has no significant change in cell number was observed over time following loss of Rnd3 expression.

This sentence has been removed, at the request of another reviewer, since the overall conclusion for the section is stated in the next paragraph.

c. This is also reflexed in our data has we observed that inhibiting or knocking down ROCK1

Corrected to- "This is also reflected in our data has we observed that inhibiting or knocking down ROCK1"

2. Its isn't clear what the rationale is behind letting A549s invasion assays being done for 24 hours, whereas H460s are done for 48 hours? Kindly explain.

The basal rates of invasion for both cell lines differ. Following our optimization experiments we achieved ~50% confluency of the invasion membrane for control A549 cells at 24h and H460 cells at 48h. These conditions were then maintained throughout the following experiments has this ~50% confluency of control cells allows us to observe either an increase or decrease in invasion phenotype, following treatments. This information has been added to the methods section and in the main text.

3. Adding molecular weight marking for all protein western blots is mandatory and there is no excuse for not included this in the blot images.

Molecular weights have been added to all Western blots

4. Please also add scale bars to microscopy images.

Scale bars have been added

5. Was DMSO used as the control for Y27632 treatment? Please clarify in the experimental protocol and when discussion the data.

Yes, DMSO was used as the control for Y27632 treatment, we have clarified this in both the method section and when discussing the data.

Major remarks-

1. Knockdown studies are integral to the interpretations made and lack of quantitative evaluation of knockdowns with biological replicates is needed. Will be useful to share KD for all the sets of experiments used for migration and other functional studies. Quantitative scatter plots for WBs should be added.

We have performed and included densitometry on all Western blotting data associated with this manuscript this includes independent biological replicates from samples used in function assays, validating either knockdowns or changes in protein expression. We have included all the corresponding plots and statistical significance in the main figures were possible or alternatively in supplemental figures. All previous claims of knockdown, decreased, increased, and no change in protein expression made in the original manuscript still stand and are now supported by the densitometric analysis. We can confirm that following siRNA knockdown of specific proteins we observed a significant decrease in that protein's expression compared to control cells.

2. Reconstitution with siRNA resistant mutants to evaluate major functional outcomes affected by knockdo3n are restored will be useful to show.

These experiments would required substantial time and resources. It would require time to perform optimization of the expression vector and generation of stable cell lines, as transient transfection rates with

plasmids in these cell lines is below 70%. We believe data generated using multiple siRNA oligos in multiple cell lines is strong enough to support the main conclusions in the manuscript without performing reconstitution studies.

3. Please add images for experiments done to evaluate cell migration by performing wound healing assays. We have added representative wound healing images of A549 and H460 control and Rnd3 knockdown cells at 0 h and assay end point to Figure 3K and have added more details on how measurements were taken to the methods section. To conserve space, we have not included other representative images of wound healing assays as the plots more accurately reflect the data.

4. Does Y27632 inhibitor treatment affect the percentage invasion of cells in Rnd3 knockdown cells? No significant difference in invasion rate was observed in Rnd3 knockdown cells treated with Y-27632 compared to DMSO treatment. When comparing the invasion rate of Rnd3 knockdown cells treated with either DMSO or Y-27632, no significant difference in invasion was observed for either Rnd3 oligo or cell line (A549 siRnd3 oligo A $p = > 0.999$, oligo D $p = 0.761$, H460 siRnd3 oligo A $p = 0.998$, oligo D $p = 0.995$) (Figure 4E-F). The same pattern was observed for cell migration in Rnd3 knockdown A549 cells treated with either DMSO or Y-27632 having no significant difference in migration rate ($p = 0.640$) (Figure 4J). We have added this data to the results section.

5. In evaluating the Rnd3 mediated regulation of ROCK1/2, autophosphorylation of ROCK1 at Ser1333 and of ROCK2 at Ser1366 could be evaluated to check their activation status in cells. We have not checked phosphorylation status of ROCK1/2. The main mechanism in which ROCK1 is known to effect cell migration and invasion in fibroblast and melanoma cells is through its regulation of actin stress fiber turnover. As we see no increase in either actin stress fibers or ROCK1 downstream signaling (no increase in phosphorylation of MLC2 or Cofilin) in Rnd3 knockdown A549 or H460 cells we did not dissect this pathway additionally to investigate the autophosphorylation status of ROCK1/2. We have modified our language in the manuscript from "ROCK1 activity" to "ROCK1 signaling". It would be interesting to test if the autophosphorylation status is altered by loss of Rnd3. However, we suspect that Rnd3 is functioning via an alternative pathway to ROCK1, as we don't see an increase in actin stress fibers and neither ROCK1/2 inhibition with Y-27632 treatment nor ROCK1 or ROCK2 double knockdown could rescue the migration or invasion rates of Rnd3 knockdown lung adenocarcinoma or PDLBM cell lines.

6. Image quality and resolution to show actin stress fibres in Fig. 4 B, C is not very clear. Also, in image 8 B, the image quality is poor. The localization of alpha 5 is not seen at the expected places. Will be useful to show insets, or better-quality zoomed in images.

We apologize for the low-quality images. We have included larger, high-resolution images of actin staining in A549 and H460 control and Rnd3 knockdown cells treated with either DMSO control or the Y-27632 ROCK1/2 inhibitor in Figure 4. We have also added additional actin staining images in supplemental Figure S2C-D.

We have now included large high-resolution images of the Alpha 5 staining, making it possible to see both puncta as well as diffused staining in the cytoplasm, which is reminiscent of alpha 5 staining. For validation of the Alpha 5 staining, we have included images and quantification of Alpha 5 fluorescent intensity in control, siRnd3, siAlpha 5 and siAlpha5:Rnd3 double knockdown A549 cells. These data show both a significant increase in Alpha 5 staining in Rnd3 knock down cells (Figure 8A-B, S8D). Importantly, we also observed a significant decrease in Alpha 5 staining compared to NSC cells in conditions in which Alpha 5 has been knocked down, siAlpha5 and siRnd3:Alpha5 (Figure 8A-B), indicating the staining observed in NSC and Rnd3 knockdown cells is that of Alpha 5 staining. This information has been added to the main text in the manuscript.

7. Quantitation of the actin stress fibres is needed to make any comment on the number of stress fibres in Y27632 inhibitor untreated/ treated cells.

To bolster our comments about the number of stress fibers we have quantified the number of stress fibers in both A549 and H460 control cells and Rnd3 knockdown cells with and without treatment with the ROCK1/2 inhibitor, Y-27632. We observed robust changes to actin stress fiber number, for example significant reduction in stress fibers following Y-27632 treatment. To quantify stress fiber number per cells, we imaged 5 fields of view (FOV) using a 60x oil objective and counted the number of stress fibers in 5 cells per FOV for each condition, over 3 biological repeats, average stress fiber number per cell was calculated for each condition per biological repeat and was plotted mean \pm standard error of the mean (SEM), one-way ANOVA with Dunnett's multiple test correction compared to NSC was performed. This information has been added to the main text, methods, and in Figures 4 and S2.

8. "Worm-like" actin structure is not a standard terminology. It will help to discuss these structures in detail, and highlight also cross- reference with literature.

We have removed this observation from the manuscript as we do not go into detail about what these structures are. From what we have observed so far, these structures don't fall into the category of known actin structures such as filopodia or invadopodia we would like to perform a complete classification of these structures, which falls outside of the scope of this study.

9. Rnd3 knockdown causes a decrease in integrin beta 3 and increase in integrin alpha 5 integrin. While the authors have further established the role of integrin alpha 5 in regulating Rnd3 mediated migration and invasion, they have not commented on the possible role of integrin beta 3. Its possible implications are useful to comment on

The expression level of beta 3 integrin is significantly decreased in the Rnd3 knockdown A549 cells compared to NSC, however we have not investigated how this influences migration or invasion rates of these cells. This loss of expression could be due to a compensation mechanism or altered integrin heterodimer composition of the cell resulting in decreased beta 3 expression however further investigation is required before we can speculate on beta 3's involvement in this pathway. However, the fact we see alterations in other integrins suggests the consequences of knocking down Rnd3 have wider implications. Here we chose to focus on alpha 5 as we see a clear rescue of the invasion and migration phenotype, we will investigate beta 3's involvement as well as profile other integrins for their involvement in this mechanism in future studies. We have included a line in the discussion about beta 3 data.

10. The discussion section is a bit lengthy and can be made crisper and to the point. The authors could consider discussing the possible mechanistic involvement and role for $\alpha 5$ regulation (surface vs total, transcription vs trafficking) in regulating Rnd3 mediated migration and invasion in LUAD cell lines.

We have streamlined the discussion and have elaborated on possible mechanisms of the Rnd3-Alpha 5 pathway maybe regulating cell migration and invasion in lung adenocarcinoma cells, in the penultimate paragraph in the discussion.

Reviewer #2 (Comments to the Authors (Required)):

This is an interesting manuscript which builds on previous publications showing the contribution of Rnd3 to cancer migration and invasion and that higher Rnd3 expression correlates with worse prognosis in several cancer types, including lung cancer as further reported here. The authors find that lung cancer-derived cell lines depleted of Rnd3 have higher $\alpha 5$ integrin expression, and that decreased $\alpha 5$ integrin expression reverses the effects of Rnd3 depletion on migration and invasion using standard in vitro models. This extends on a previous study reporting that Rnd3 regulates integrin expression. The manuscript does not explain the molecular basis for this integrin upregulation, which could be investigated in the future.

General interpretation. Rnd3 is known to be phosphorylated by ROCK1, which inhibits its activity, but whether it signals through inhibiting Rho/ROCK signaling is not consistently established across different models, nor is the molecular basis for any effect fully understood. The authors should consider their wording on this topic carefully, giving both sides of the argument.

1. Figure 1. Explain the cut off for Rnd3 high/low expression in patients -what cut off was chosen? This appears to have been chosen to be optimal for observing an effect. What happens if you use different cut-offs or just use the patients with the highest expression and lowest expression (e.g. top 25% and bottom 25%)? Add a graph showing the range of Rnd3 expression in the patients and where the cut-off is. This appears to be replicating previous data - was that using a different set of patient samples? More information is needed in the text and a new graph.

We apologize for the lack of information and clarity. The original cutoff groups for high and low Rnd3 expression were generated using the 'survminer' R package to determine the optimal cutpoint for Rnd3 expression; this package uses the maximally selected rank statistics from the 'maxstat' R package. This package finds the most statistically significant cutoff and is commonly used when identifying survival signatures as it utilizes all of the data set. We have now also included an analysis using the top and bottom quartiles and generated Kaplan-Meier plots for these groups, we observed patients with low levels of Rnd3 have significantly higher survival probability ($p = 0.0048$) (Figure S1A). We have also included the risk tables for each of the Kaplan-Meier plots (Figure 1, S1A), as well as the expression range plot denoting the cut-points for the groups utilized in the survival analysis (Figure S1B).

Although Li et al 2016 utilized the TCGA database their analysis is distinct from ours: They compared Rnd3 expression in normal lung tissue samples to squamous cell carcinoma and lung adenocarcinoma samples. Their high vs low Rnd3 survival analysis used a none specified sample set of 2437 lung cancer cases and 2435 controls, this may or may not contain the TCGA LUAD data set (comprising of 701 samples, down to 501 once matched normal samples have been removed) which we used for our analysis. However, we only utilized the TCGA LUAD data set for our analysis to investigate if Rnd3 expression levels correlated with

patient outcome in lung adenocarcinoma specifically, our data is in line with other groups including Li et al which observe high Rnd3 expression correlates with poor patient outcome but have performed their analysis on different cohorts of lung cancer patient samples. We have made clear that our analysis is unique but is inline with other similar studies.

This additional information and data have been added to the manuscript.

2. It is not surprising that Rnd3 depletion decreases migration and invasion of the cell lines used in this study. This has been demonstrated in multiple studies over many years in a wide variety of cell lines (or that increased expression increases invasion). The type of cell lines used is not important because they are all cultured in vitro on hard plastic surfaces. All that differs for the two brain metastasis lines is possibly that that they have been cultured for less time in vitro than the two long-standing lung adenocarcinoma cell lines used. The authors should state specifically that these data are consistent with many other studies.

We have stated that our data, Rnd3 depletion decreases migration and invasion in lung adenocarcinoma cells, is consistent with other studies in other cell lines, in both the results and discussion.

3. The effect of Rnd3 depletion on cell proliferation/cell cycle progression depends on the cell line. It is not really surprising that highly proliferative cancer cells are not affected by Rnd3 depletion in these assays, because they are already likely to have mutations that bypass regulation of cell cycle progression. Please comment on whether these lines have such mutations e.g. by checking DEPMAP portal of cancer cell lines. We agree with the reviewer, and suspect that background mutations especially in cell lines, may override Rnd3's effect on proliferation. A549 and H460 cells have KRAS mutations which may drive the proliferation of these cells regardless of Rnd3 expression levels.

We have added a line about Rnd3 effects on proliferation maybe due to background mutations: "...potentially dependent on the mutational background of the cell." As well as information about the mutational background of A549 and H460 cells "A determining factor for whether altering Rnd3 expression levels effect cell proliferation could be dependent on background mutations present in cell lines, these could bypass Rnd3's effect on proliferation. Both A549 and H460 cells have KRAS mutations, which could drive cell proliferation and survival independently of Rnd3 expression levels."

4. Figure 2A: provide a graph quantifying Rnd3 protein levels over at least 3 independent experiments.

We can confirm for Figure 2 that following siRNA knockdown of Rnd3 at each time point we observed a significant decrease in Rnd3 expression compared to control cells (Figure S1).

Additionally, we have performed and included densitometry on all Western blotting data associated with this manuscript this includes independent biological replicates from samples used in function assays, validating either knockdowns or changes in protein expression. We have included all the corresponding plots and statistical significance in the main figures were possible or alternatively in supplemental figures. All previous claims of knockdown, decreased, increased, and no change in protein expression made in the original manuscript still stand and are now supported by the densitometric analysis.

5. For all graphs, provide absolute p values so that readers can judge for themselves the level of significance. This is considered best practice now.

We have opted to keep the asterisks to represent significance on the plots as we feel this is visually more intuitive for the reader and is in line with LSA guidelines.

6. Figure 4 and Figure S1. To prove that Y27632 is active in the author's assays, biochemical evidence is required. Cofilin phosphorylation can be mediated by multiple kinases and is not a ROCK-specific target. pMLC changes are also indirect because ROCKs actually phosphorylate pMYPT, which should be tested as well to be sure that the ROCK inhibitor is working correctly (and whether Rnd3 knockdown alters ROCK1 signaling. Before concluding that it has no effect on invasion (which is surprising) the authors need to demonstrate the inhibitor is active at the concentration they use. They should also titrate Y27632 to check whether higher concentrations affect invasion (alone, irrespective of +/- Rnd3).

Regarding whether Rnd3 knockdown alters ROCK1 signaling in A549 or H460 cells our data suggests it does not. As the reviewer states Cofilin is not a direct target of ROCK1, activated ROCK1 phosphorylates LIMK1/2 which in turn then phosphorylates cofilin to prevent actin stress fiber disassembly. ROCK1 does directly phosphorylate MLC2 at Ser19 to regulate actin stress fiber assembly, and although this phospho-site can be phosphorylated by other kinases including ROCK2 we see neither an increase in MLC2 Ser19 phosphorylation nor indirect Cofilin phosphorylation in Rnd3 knockdown cells. Additionally, we did not observe an increase in actin stress fibers in Rnd3 depleted A549 or H460 cells, adding to the evidence that ROCK1 signaling is not increased in Rnd3 knockdown A549 or H460 cells. We have added the following text to clarify this information in the manuscript. "Surprisingly, no increase in ROCK1 signaling was observed in either Rnd3 knocked down A549 or H460 cells, as we did not observe either an increased actin stress fibers or increased phosphorylation of either direct or indirect ROCK1 downstream targets, MLC2 or cofilin, compared to NSC cells."

Regarding whether the ROCK1/2 inhibitor Y-27632 is working correctly, we believe that the Y-27632 inhibitor is active and functioning to inhibit ROCK1. We observe a significant decrease in the number of actin stress fibers, to near complete abolishment of actin stress fiber in both A549 and H460 NSC cells as well as Rnd3 knockdown cells following 5 μ M treatment of Y-27632. We have included high-resolution images and quantification of actin stress fiber (**Figure 4A-D**). This loss of actin stress fibers is indicative of ROCK1 inhibition following treatment with Y-27632. Additionally, we also observe a significant increase ($p = 0.0103$) in cell invasion in NSC H460 cells treated with 5 μ M Y-27632 compared to DMSO-treated NSC H460 cells (**Figure 4F, H**). It is only the A549 cells in which 5 μ M Y-27632 had no significant effect on the invasion rates of NSC cells compared the DMSO-treated NSC A549 cells, there is a slight increase in both invasion and migration following Y-27632 treatment, but it is not significant (**Figure 4E, G, I-J**). As requested, we performed a titration of the Y-27632 treatment in both A549 and H460 cells. We compared 5 μ M, the common working concentration of Y-27632, to 10 μ M of Y-27632 with the equivalent volume of DMSO as controls. Increasing the concentration of the Y-27632 inhibitor pass 10 μ M leads to additional off target effects through the inhibition of addition kinases making data difficult to interpret. First, we quantified the number of actin stress fibers in non-transfected A549 and H460 cells treated with either 5 μ M or 10 μ M of Y-27632 compared to their corresponding DMSO control treatments. Similar to data generated in the NSC A549 and H460 cells, we observed a significant decrease in the number of actin stress fibers, to almost complete ablation, in both non-transfected A549 and H460 cells following 5 μ M treatment of Y-27632, and observed no further decrease in the number of stress fibers following 10 μ M treatment, indicating that stress fibers were abolished at 5 μ M and doubling the concentration of Y-27632 had no additional effect on reducing stress fiber number (**Figure S2C-F**). We also performed invasion assays on A549 and H460 cells treated with either 5 μ M or 10 μ M of Y-27632, and their corresponding DMSO control treatments. We again observed no increase in invasion in A549 cells treated with either 5 μ M or 10 μ M of Y-27632, we did observe a significant increase H460 invasions rates following Y-27632 treatment however doubling the concentration of Y-27632 had no significant effect on the invasion rate compared to the standard 5 μ M treatment (**Figure S2G-H**). With the significant loss of actin stress fibers in both A549 and H460 cells, and the significant increase in cell invasion in H460 cells following 5 μ M of Y-27632 we conclude that Y-27632 is active and functioning to inhibit ROCK1 in these cell lines. This information and data have been added to the manuscript.

7. The single images shown in Figure 4 +/- Y27632 and Rnd3 depletion are very difficult to interpret, since the lung adenocarcinoma cell lines have few visible stress fibers (not surprising as most cancer cell lines have few stress fibers). Cell shape should be quantified instead to determine of Y27632 and/or Rnd3 depletion are having expected effects.

We have added larger high-resolution images and have also quantified the number of stress fibers in both A549 and H460 control cells and Rnd3 knockdown cells with and without 5 μ M treatment with the ROCK1/2 inhibitor, Y-27632. Although lung adenocarcinoma cell lines do have fewer stress fibers compared to fibroblast, we are able to observe and quantify both the presence and loss of stress fibers in these cell lines so we opted for this metric. We also observed a change in cell shape following Y-27632 treatment, but we have not quantified that. To quantify stress fiber number per cell, we imaged 5 fields of view (FOV) using a 60x oil objective and counted the number of stress fibers in 5 cells per FOV for each condition, over 3 biological repeats, average stress fiber number per cell was calculated for each condition per biological repeat and was plotted mean \pm standard error of the mean (SEM), one-way ANOVA with Dunnett's multiple test correction compared to NSC was performed. We observed either no change or a significant reduction in actin stress fiber number in Rnd3 knockdown cells compared to NSC cells, and a significant reduction in actin stress fibers, almost to zero, following Y-27632 treatment. This information has also been added to the main text, methods, and in Figures 4 and S2.

8. Figure 5. The authors should change the wording concerning the effects of knocking down ROCK1 and ROCK2 on the Rnd3 migration/invasion phenotypes, given that ROCK1/2 knockdown is not very strong yet they observe a significant effect in one of the two lung cancer cell lines tested. They cannot conclude that the Rnd3 knockdown phenotype is independent of ROCK1/2 given these data, but that it acts partially through ROCK1/2. See also point above that they have not conclusively proven that Y27632 is active in their experiments.

Regarding Y-27632 treatment and inhibition of ROCK1, we have added additional data and high-resolution images to support that Y-27632 treatment is indeed inhibiting ROCK1 our experiments, see above comments. We have also included direct statistical comparison between Rnd3 depletion and joint Rnd3/ROCK1 or ROCK2 depletion, in response to your next comment (9), which highlights that neither knocking down ROCK1 nor ROCK2 in Rnd3 depleted cells significantly affected the invasion or migration rates of Rnd3 knockdown A549, H460, or the PDLBM cells.

We have performed densitometric analysis on all the Western blotting data present in the manuscript and observe a significant decrease in protein expression compared to NSC levels following siRNA knockdowns, validating the knockdowns.

We have changed the wording in the text and Figure 5 legend stating that “Knockdown of ROCK1 or ROCK2 does not fully rescue the migration or invasion phenotype of Rnd3 knockdown in A549 or H460 lung adenocarcinoma”. Due to the observation that compared to NSC there is a partial rescue of H460 cells in the Rnd3:ROCK2 doubles. However, this is Rnd3:ROCK2 doubles and Rnd3 single knockdown cell invasion rates are not significantly different (siRnd3 compared to siRnd3:ROCK1 $p = 0.910$, or siRnd3:ROCK2 $p = 0.237$). With us adding the additional data and more clarity, we hope we provided enough evidence to support our claim that Rnd3 is functioning via alternative pathway to ROCK1 to regulate cell migration and invasion in lung adenocarcinoma cells.

9. Graphs in Figures 5 and 6 do not show the direct statistical comparison between Rnd3 depletion and joint Rnd3/ROCK1 or ROCK2 depletion, and in Figure 7 between Rnd3 depletion and Rnd3/RhoA. The same holds for supplemental figure graphs. These seem the most relevant comparisons. In any case, tables showing all the statistical comparisons (every condition against every other condition) should be provided with absolute p values. The authors will have these data from the ANOVA tests.

We agree with the reviewer that these are indeed important comparisons which bolster our conclusions. All the above-mentioned comparisons were not statistically significant which is why we did not include them on the plots. We have rectified this oversight of not commenting on these comparisons and have now described them in their relevant results sections:

When comparing the invasion or migration rates of Rnd3 knockdown to siRnd3:ROCK1 or Rnd3:ROCK2 double knockdown in A549 cells, no significant change in invasion rate was observed (siRnd3 compared to siRnd3:ROCK1 $p = 0.972$, or siRnd3:ROCK2 $p = 0.963$) or migration rate (siRnd3 compared to siRnd3:ROCK1 $p = 0.681$, or siRnd3:ROCK2 $p = 0.988$). Similarly, when comparing the invasion rates of Rnd3 knockdown to siRnd3:ROCK1 or Rnd3:ROCK2 double knockdown in H460 cells no significant change in invasion rate was observed (siRnd3 compared to siRnd3:ROCK1 $p = 0.910$, or siRnd3:ROCK2 $p = 0.237$). These data suggest, along with the comparison with NSC cells, that knocking down either ROCK1 or ROCK2 in either A549 or H460 cells in which Rnd3 is knocked down, has no effect on cell migration or invasion.

When comparing the invasion rates of Rnd3 knockdown to siRnd3:ROCK1 double knockdown in 601 cells no significant change in invasion rate was observed (siRnd3 compared to siRnd3:ROCK1 $p = 0.899$). Similarly, when comparing the invasion rates of Rnd3 knockdown to siRnd3:ROCK1 double knockdown in 620 cells no significant change in invasion rate was observed (siRnd3 compared to siRnd3:ROCK1 $p = 0.997$). These data suggest along with the comparison with NSC cells, that knocking down ROCK1 in either PDLBM cell line in which Rnd3 is knocked down, has no effect on cell invasion rates.

Comparison of Rnd3 single knockdowns compared to Rnd3:RhoA double knockdowns was not significantly different in either cell migration rates in A549 cells (siRnd3 compared to siRnd3:RhoA(1) $p = 0.999$, or siRnd3:RhoA(2) $p = 0.710$) or cell invasion rates for H460 cells (siRnd3 compared to siRnd3:RhoA(1) $p = 0.327$, or siRnd3:RhoA(2) $p = >0.999$). These data suggest along with the comparison with NSC cells, that knocking down either RhoA in either A549 or H460 cells in which Rnd3 is knocked down, has no effect on cell migration or invasion.

10. Figure 7: the authors should interpret these results with caution because RhoC can also activate ROCK1/ROCK2 and could be the most important Rho subfamily member in these cell lines. It is clear from several studies that RhoC depletion strongly reduces invasion and metastasis, whereas the effect of RhoA depletion is variable depending on the cell type. Moreover, since depletion of RhoA in these cells reduces cell migration, how would they expect it to rescue the effect of Rnd3 depletion on cell migration? It should be synergistic in decreasing migration, if anything.

It would be interesting to investigate RhoC in the context of Rnd3 knockdowns in lung cancer cells and we will look into it. We investigated RhoA 1) because of its ability to activate ROCK1 and 2) that loss of Rnd3 may increase RhoA signaling, when Rnd3 is expressed, it can recruit p190RhoGAP to RhoA inhibiting RhoA signaling. Making it a logical target. We were not sure what the outcome of knocking down RhoA alone or in combination with Rnd3 would be in either A549 or H460 cells, as the reviewer points out RhoA's effect on cell migration and invasion is variable depending on cell type. We were expecting an increase in cell migration and invasion following RhoA knockdown, and thus performing the double knockdown with Rnd3 would make more sense. This is not what we observed, and the conclusions we gathered from this data is that loss of RhoA expression in A549 and H460 cells decreases cell migration and invasion, respectively, and double knockdown with Rnd3 did not rescue migration or invasion rates. We have clarified this in the discussion.

11. Figure S2 requires quantification of western blots for each integrin over multiple experiments to demonstrate reproducibility of the results.

We have performed densitometry on integrin expression over multiple biological repeats and can confirm for Figure S2, now Figure S7 and S8, that 1) Following siRNA knockdown of Rnd3 in each sample we observed a significant decrease in Rnd3 expression compared to control cells, and 2) We observed no change in Beta 1,

Beta 5, Alpha V, but did observe a significant decrease in Beta 3 expression in Rnd3 knockdown A549 cells compared to NSC cells. All Alpha 5 densitometry have been moved to Figures 8, S7 and S8.

Additionally, we have performed and included densitometry on all Western blotting data associated with this manuscript this includes independent biological replicates from samples used in function assays, validating either knockdowns or changes in protein expression. We have included all the corresponding plots and statistical significance in the main figures were possible or alternatively in supplemental figures. All previous claims of knockdown, decreased, increased, and no change in protein expression made in the original manuscript still stand and are now supported by the densitometric analysis.

Minor points:

1. There are grammatical errors throughout the manuscript that need to be corrected.

We have corrected grammatical errors

2. There are inconsistencies in how 'hours' are abbreviated throughout. The accepted abbreviation is h (not hrs).

We have changed all abbreviated 'hours' to h throughout.

3. Scale bars are missing and need to be added to all images.

Scale bars have been added to the images.

4. Figure 3, give absolute n values for each condition rather than >4.

Absolute n values for each condition have been added to Figure 3.

5. Y27632 inhibits ROCK1, ROCK2 and PRKs (see papers testing multiple kinase inhibitors on multiple kinases from the Cohen laboratory in Biochemical Journal). This should be stated in the text and every time it is used, state that it is a at least a ROCK1/2 inhibitor (not just ROCK1).

We agree that this is a very important point, we have edited the text although to be clear that Y-27632 inhibits ROCK1, ROCK2 and PRK2 at the common working concentration 5 and 10 μ M. We have also added the additional reference from the Cohen lab and state Y-27632 is a ROCK1/2 inhibitor.

Reviewer #3 (Comments to the Authors (Required)):

In this work, Garcia-Garcia et al. show that Rnd3 high expression correlates with a worse prognosis in lung cancer patients. They showed that Rnd3 mediates cell migration and invasion in 2 different cell lines of lung adenocarcinoma without significant effects on proliferation or viability. Since Rnd3 inhibits ROCK1/2 signalling, they went on to test whether Rnd3 could be mediating cell migration and invasion via this pathway. Inhibiting or silencing ROCK1 had no effect on the invasion or migration, and thus it seems like Rnd3 is controlling cell migration and invasion through a different pathway. Finally, the authors show that there is an upregulation of alpha5-integrin upon Rnd3 silencing and this upregulation prevents the cells from migrating/invading. Since in the literature there are reports of Rnd3 being either a tumor suppressor or tumor promoter, this work becomes relevant and the use of patient-derived metastatic cells reinforces the main point. Overall, the study is conducted properly and data supports the main conclusion; however, some aspects of the mechanism, discussion and the results presentation need to be improved.

1) The data strongly supports a role for Rnd3 in cell migration and invasion in lung adenocarcinoma.

2) The data strongly shows that Rnd3 mediates LUAD migration and invasion through a ROCK-independent pathway.

3) The data suggests that Rnd3 might repress expression of integrin alpha5, and that alpha5-integrin expression prevents LUAD cell migration/invasion. This is surprising since alpha5-integrin is known for mediating cancer cell invasion in other cancers. Moreover, the staining shown in the results is difficult to see and looks very unusual. The authors need to replace the IF picture of alpha5-integrin staining for a higher resolution picture to show localization or absence of alpha5-integrins at adhesion sites and/or cell-cell contact sites and validate the specificity of the staining against alpha5-knockdown cells. Moreover, they should indicate their rationale for looking at the integrins they are showing and include quantification of the blots and data showing expression of alpha6 and alpha2 to understand if the effect is specific to alpha5.

4) Specific points to address regarding the results, text and figures:

-In the introduction, 3rd paragraph, page 2, the authors mention that "ROCK1, in turn, can phosphorylate Rnd3, increasing Rnd3 levels and producing a negative feedback loop(Riou et al., 2013)." Here, the authors

should also reference Komander 2008, EMBO J, doi: 10.1038/emboj.2008.226

We apologize for our oversight and have now included the reference.

-Please correct typos in page 3: "Rnd3 has been proposed to regulate many different cellular processes that are commonly dysregulated in cancer" and "These data suggest that Rnd3 may regulate the metastatic potential..."

We have corrected the typos. "Proposed" to "proposed" and "regulated" to "regulate"

-Add "that" in the following sentence: "We utilized patient derived lung-to brain metastasis (PDLBM) cells as a proxy for advanced disseminated disease and observed that knocking down Rnd3 expression in these cell lines resulted in a significant decrease in cell invasion"

We have added "that" to the above sentence in the manuscript.

-In page 4, this sentence should be deleted since the overall conclusion for the section is stated in the next paragraph: "These data indicate that depletion of Rnd3 does not affect cell viability or proliferation has no significant change in cell number was observed over time following loss of Rnd3 expression."

We agree and have removed this sentence.

-In the discussion, please mention that the studies showing a role for Rnd3 in proliferation have been done in fibroblasts or other cell types, and not in LUAD cells to help the reader put the previous work in context.

We have added the following statement "Both overexpression and decreased Rnd3 expression have been reported to affect cell cycle progression, through a variety of different mechanisms, in different cell types, cancer types, and potentially dependent on the mutational background of the cell. Most of the studies investigating Rnd3's role in proliferation have not been performed in lung cancer cells". We have also edited the paragraph to clarify which cell lines were used to study Rnd3's role in proliferation.

-References: DOI accession number for Wennerberg, K., et al. (2003), is incomplete. Please add the correct DOI link.

We have now added the correct DOI number

-The authors mention in the manuscript that they only looked at a small panel of integrins. What is the rationale for choosing those integrins? What is the composition of the extracellular matrix in the lungs? Which integrins would you expect to have in normal tissue and do those relate to the ones you looked at? If expressed in lung cells, the authors need to check alpha2-integrin and alpha6-integrin since those are the major collagen and laminin receptors.

The reviewer makes a good point and we agree that we have not fully investigated the involvement all the integrins and we intend to do this in future studies to fully understand this novel pathway we have identified here. We started with the small panel to include integrins which were both expressed and have also been implicated in regulating cell migration, we were specifically interested in beta 1, as previous work by Endzhievskaya et al, has shown a link to Rnd3 and beta 1 activation. We also wanted to investigate beta 1's common heterodimer partner, integrin alpha 5, which was not investigated by Endzhievskaya et al, as the alpha5:beta1 heterodimer has been implicated in regulating cell migration and invasion. From our limited panel we observed an increase in alpha 5 and decrease in B3, for this study we opted to focus on alpha 5, which we establish is involved in Rnd3 regulation of cell migration and invasion in lung cancer cells. However, to fully understand the mechanism we agree that our future focus must include investigating the involvement of beta 3 and other integrins not tested here. We also need to establish the mechanism by which Rnd3 is regulating the expression of alpha 5 and beta 3 as well as how this results in decreased cell migration and invasion. Some possible mechanisms could include increased or decreased cell adhesion to the ECM and disruption of adhesion assembly and turnover.

-Please check and rephrase the second paragraph of the discussion (page 8). It contains some grammatical errors that make it difficult to read.

For example, the first sentence stating: "To investigate the potential mechanistic basis by which patients with low Rnd3 expression experience improved survival outcomes" makes the reader think that you would follow up patient outcomes and relate that data with tumor-derived organoids in real time, but this is not the case. A more suitable way to say this would be: "To investigate whether Rnd3 confers metastatic properties to lung

cancer cells..." or ..."whether Rnd3 controls metastasis and cell invasion".

Also, at the end of the paragraph, please replace the word "suspectable" for "susceptible" if that is what the authors intended to say.

We have replaced the first sentence with the suggested sentence "To investigate whether Rnd3 confers metastatic properties to lung cancer cells..", replaced "suspectable" for "susceptible" and corrected the grammatical errors in this paragraph.

-In the discussion (page 9, 2nd paragraph), the authors state that "Surprisingly, no increase in ROCK1 activity was observed..." However, they did not measure ROCK1 activity, but rather ROCK1 signalling. Please change the wording throughout this paragraph, and also in Figure S1.

We have altered the wording throughout and in Figure S1 (now Figure S3) from "ROCK1 activity" to "ROCK1 signaling".

-Moreover, the first part of this paragraph (page 9, 2nd paragraph) sounds repetitive (all these facts were already mentioned in the introduction). Rather than listing these facts, the authors should compare their results to what is in the literature: previous studies have been done in melanoma or fibroblasts, and here they report a different mechanism for lung adenocarcinoma. What does this mean in terms of cell migration and attachment to the extracellular matrix of the invading tissue? Are cells less/more contractile to colonize a softer tissue when they have high/low Rnd3? How is the brain ECM different to lung ECM? Is there anything known about Rnd3 blocking fibronectin deposition? What would be interesting to look at next? If the mechanism is ROCK independent, could the authors speculate on what other mechanisms could be mediating the invasive phenotype?

We have added more discussion and clarification to this paragraph and compared our data to other findings in previous literature. We have added discussion of possible mechanisms and future directions to the penultimate paragraph in the discussion section.

-For all figures, the authors should state if the western blots showed are representative of n-number of repeats. Moreover, whenever possible, the blots showed to validate silencing should be shown after the assay results.

We have added the n number for representative Western blots along with the densitometric analysis and placed the blots after the results were possible, we have also included densitometry for all Western blots and provided this data in the main figures were possible or in supplemental figures and added n-number, to aid the reader in assessing the validity of the data.

-For Figure 3 and related text in the results, the authors should mention why did they use different timepoints for different cell lines? For panel C and D, please indicate that this number is per Field of View (FOV).

The basal rates of invasion for both cell lines differ. Following our optimization experiments we achieved ~50% confluency of the invasion membrane for control A549 cells at 24 h and H460 cells at 48 h. These conditions were then maintained throughout has this ~50% confluency of control cells allows us to observe either an increase or decrease in invasion phenotype following treatments.

Similarly, both A549 and H460 have differing basal rates of cell migration. When optimizing our wound healing assays, we observed A549 fully close their wound at ~30 h and H460 at ~60 h. Following our optimization experiments, A549 and H460 control cells reach ~60% wound closure at 24 h and 48 h respectively. These conditions were then maintained throughout, has this ~60% closure of control cells allows us to observe either an increase or decrease in migration rate following treatments. This information has been added to the Methods section and in the main text.

For Figure 3D-E, we have clarified how the counting of invaded cells was performed regarding the FOV in both the method section and figure legend.

-Figure 5: the title of this figure is very long, and while it is very nice to show that you have reproduced the published results in melanoma cells (which also demonstrates the effect of the ROCK inhibitor), this could be skipped in the Figure title and only state that Rnd3 controls cell migration and invasion in a ROCK-independent fashion in LUAD cells. In addition, could the authors please reorder the results, showing the LUAD results first, and then the results from the melanoma cells, and indicate in the legend that this was used as a control?

We have renamed figure 5 to..." Knockdown of ROCK1 or ROCK2 does not fully rescue the migration or invasion phenotype of Rnd3 knockdown in A549 or H460 lung adenocarcinoma". We have also reordered the Figure putting the LUAD data first and stated in the figure legend that the Melanoma cell line data is a control.

-Figure 8: The IF staining for alpha 5-integrin seems a bit strange, almost absent from focal adhesions and more localized to cell-cell contact sites. Is this correct? Could the authors please provide a higher resolution

picture and validate their alpha 5-integrin staining against alpha5-integrin knockdown cells? Could they zoom into an area to show that alpha5-integrin is (or is not) at focal adhesion sites?

We apologize for the low-quality images. We have now included large high-resolution images of the Alpha 5 staining, making it possible to see both puncta as well as diffused staining in the cytoplasm, which is reminiscent of alpha 5 staining. For validation of the Alpha 5 staining, we have included images and quantification of Alpha 5 fluorescent intensity in control, siRnd3, siAlpha 5 and siAlpha5:Rnd3 double knockdown A549 cells. These data show both a significant increase in Alpha 5 staining in Rnd3 knock down cells (Figure 8A-B, S8D). Importantly, we also observed a significant decrease in Alpha 5 staining compared to NSC cells in conditions in which Alpha 5 has been knocked down, siAlpha5 and siRnd3:Alpha5 (Figure 8A-B), indicating the staining observed in NSC and Rnd3 knockdown cells is that of Alpha 5 staining. This information has been added to the main text in the manuscript.

The localization of the Alpha 5 staining, what Alpha 5 is colocalizing with, and how this is altered following Rnd3 knockdown is something we very interested in. This data along with profiling if other integrins play role in the process and detailing the mechanism of how the Rnd3-alpha 5 pathway is controlling cell migration and invasion are currently being actively pursued in the lab for our next publication.

-Following up from the possible localization of alpha5 integrin at cell-cell junctions, do the Rnd3 knockdown cells cluster more than the NSC cells, similar to what has been observed in Ryan et al., 2012 (doi: <https://doi.org/10.1242/jcs.101931>)?

We have now included large high-resolution images of the Alpha 5 staining, making it possible to see both puncta as well as diffused staining in the cytoplasm, which is reminiscent of alpha 5 staining. We have not observed an increase in cell clustering in the lung adenocarcinoma cells, like we observed in the HaCaT cell line. Additionally, not shown in the manuscript, desmoplakin I/II expression was extremely low in control A549 and H460 cells, which is common in lung cancer cell line, and we did not observe an increase in desmoplakin I/II following Rnd3 knockdown in the lung cancer cells, which was in part responsible for the clustering/oming phenotype we identified in the Hacat cells. As we did not observe an increase in cell clustering or increase in desmosomal proteins this data was not included in the manuscript.

-Figures S1 and S2. Could the authors please show a quantification of the blots?

We have performed densitometry on both Figure S1 (now Figure S4) and Figure S2 (now Figure S7 and S8) over multiple biological repeats and can confirm that: 1) Following siRNA knockdown of Rnd3 in each sample we observed a significant decrease in Rnd3 expression compared to control cells. 2) For ROCK1 downstream targets we observed no significant change in phosphorylation in Rnd3 knockdown cells compared with control cells (Figure S4). 3) We observed no change in Beta 1, Beta 5, Alpha V, but did observe a significant decrease in Beta 3 expression in Rnd3 knockdown A549 cells compared to NSC cells (Figure S7). All Alpha 5 densitometry has been moved to Figures 8 and S8, and we observed a significant increase in Alpha 5 expression levels in Rnd3 knockdown A549 cells compare to NSC cells (Figure S8)

Additionally, we have performed and included densitometry on all Western blotting data associated with this manuscript this includes independent biological replicates from samples used in function assays, validating either knockdowns or changes in protein expression. We have included all the corresponding plots and statistical significance in the main figures were possible or alternatively in supplemental figures. All previous claims of knockdown, decreased, increased, and no change in protein expression made in the original manuscript still stand and are now supported by the densitometric analysis.

-For all figures, please clearly indicate that you are using siRNA. For example, as siRnd3(A), siROCK1, so that it is easier for the reader to understand the plots and figures.

We have updated the labeling of all the figures to included "si", we agree this will aid the reader.

February 2, 2026

RE: Life Science Alliance Manuscript #LSA-2025-03494-TR

Dr. Katie Rose Ryan
University of Arkansas for Medical Sciences
Department of Biochemistry and Molecular Biology
University of Arkansas for Medical Sciences
Little Rock, Arkansas 72205-7101

Dear Dr. Ryan,

Thank you for submitting your revised manuscript entitled "Rnd3 regulates lung cancer cell invasion and migration independently of ROCK1 via alpha 5 integrin".

Your manuscript has been carefully evaluated by all the original reviewers whose comments are appended below. The three reviewers are consistent in that your revised manuscript has addressed their significant concerns. Additionally, Reviewers 2 and 3 have carefully highlighted several minor points, all of which, we agree must be addressed.

Overall, we would be happy to publish your paper in Life Science Alliance pending resolution of the above points and final revisions necessary to meet our formatting guidelines.

MANUSCRIPT ORGANIZATION AND FORMATTING:

To avoid unnecessary delays in the acceptance and publication of your paper, please read the following information carefully. Full guidelines are available on our Instructions for Authors page, <https://www.life-science-alliance.org/authors>

- Please improve clarity in the title of the manuscript in particular the second part, "independently of ROCK1 via alpha 5 integrin"
- In agreement with Reviewers 2 and 3, please run a thorough grammatical check on the entire text.
- Please split the Summary Blurb into two sentences.
- Please improve clarity in the abstract by referring to "lung adenocarcinoma A549 cells" (line 44) and "patient-derived cell lines from lung to brain metastases" (line 45). Please rephrase run-on sentences in lines 46-47 and 50-53.
- Please provide an identifier or details for the TCGA-LUAD dataset stated in the methods section.
- In the methods section, for the patient-derived brain metastatic non-small cell lung carcinoma cells, please follow LSA guidelines and identify the committee approving the experiments, and include a statement that informed consent was obtained from all subjects and that the experiments conformed to the principles set out in the WMA Declaration of Helsinki and the Department of Health and Human Services Belmont Report (<https://www.life-science-alliance.org/editorial-policies#humans>).
- Please provide sequence information for oligonucleotides used in this study.
- For section on microscopy, please include details on objective (type, NA).
- Please include a brief description of imaging for cell migration assay and transwell invasion assay.
- Please provide a scale bar and size information for Figures 3B, 4B, 4D, 6B, 8B.
- Please add the X and Bluesky handles of your host institute/organization, as well as your own and/or one of the authors, in our system
- Please be sure to mention all authors in the Authors' Contributions section in the manuscript file.
- please add callouts for Figures S2A-B and S4A-I to your main manuscript text;
- Please be sure that the authorship listing and order is correct.

If you are planning a press release on your work, please inform us immediately to allow informing our production team and

scheduling a release date.

LSA encourages authors to provide a 30-60 second video where the study is briefly explained. We will use these videos on social media to promote the published paper and the presenting author (for examples, see <https://docs.google.com/document/d/1-UWCfbE4pGcDdcgzcmiuJI2XMBJnxKYeqRvLLrLSo8s/edit?usp=sharing>). Corresponding or first-authors are welcome to submit the video. Please submit only one video per manuscript. The video can be emailed to contact@life-science-alliance.org

FINAL FILES:

The following items are required for acceptance.

The license to publish form must be signed before your manuscript can be sent to production. A link to the license to publish form will be available to the corresponding author only. Please take a moment to check your funder requirements.

Thank you for your attention to these final processing requirements. Please revise and format the manuscript and upload materials as soon as you are able.

Thank you for this interesting contribution to the literature. We look forward to publishing your paper in Life Science Alliance.

Sincerely,

Sarita Hebbar, PhD
Scientific Editor
Life Science Alliance
<http://www.lsjournal.org>

Reviewer #1 (Comments to the Authors (Required)):

The revision has addressed most of concerns of the study and could be accepted for publication.

Reviewer #2 (Comments to the Authors (Required)):

The authors have included new data, information and carried out quantification in response to my comments, that answer all the points I raised satisfactorily. The manuscript is much improved with the western blot quantifications and additional data. However, there are some minor errors that need correcting in the new figures and text:

1. Figure S1: numbers of patients are incorrect in figure legend (cut and pasted from Figure 1) but correct on figure (126/501).
2. Scale bar sizes (figure legends) are in μm not μM .
3. There is ambiguity about n numbers for stress fiber quantification: Figure 4 legend C-D states n=3 for stress fiber quantification but methods section states n=4.
4. Figure 5D-F, H: relevant statistical comparisons here for migration and invasion are Rnd3 knockdown versus Rnd3 knockdown plus ROCK1 or ROCK2 knockdown, which were provided in the reply to the reviewers and have been added in the

text but not added to the graphs. These comparisons should be included in the graphs to enable readers to see the information easily.

5. Discussion on Rnd3 +/- RhoA knockdown - you should point out that the double knockdown did not further decrease migration/invasion beyond either individually. As mentioned in the reviewer comments, if anything the double knockdown should further decrease migration/invasion, rather than any rescue by RhoA knockdown.

6. Some grammatical and spelling errors in places - please check carefully prior to submission of final version.

7. Data = plural (you have sometimes put e.g. 'this data' instead of the correct 'these data').

Reviewer #3 (Comments to the Authors (Required)):

The revised version of the manuscript nicely addressed the main issues from the previous version, including proper quantifications of blots, stress fibers, high-quality images, activity of the ROCK inhibitor and the specificity of the alpha5 integrin staining. I recommend the revised version to be accepted with these additional minor corrections:

A) Quantification of actin stress fibers: the authors should state at the plot Y axis that this is the proportion of stress fibers relative to NSC cells, not the number. Otherwise, it could be interpreted as if the NSC cells had only 1 stress fiber and the silenced cells half a fiber per cell.

B) Please check carefully for grammatical errors for the final version of the manuscript. From the added text, I could mention:

-Lines 421-423: Please kindly rephrase this sentence, the point is valid and relevant but it is hard to read since it contains redundant phrasing.

-Line 462: "an increase in actin stress fibers" instead of "an increased actin stress fibers"

-Line 554: "implicated" instead of "implication"

February 10, 2026

RE: Life Science Alliance Manuscript #LSA-2025-03494-TRR

Dr. Katie Rose Ryan
University of Arkansas for Medical Sciences
Department of Biochemistry and Molecular Biology
University of Arkansas for Medical Sciences
Little Rock, Arkansas 72205-7101

Dear Dr. Ryan,

Thank you for submitting your Research Article entitled "Rnd3 regulates lung cancer cell invasion and migration independently of ROCK1 via alpha 5 integrin". It is a pleasure to let you know that your manuscript is now accepted for publication in Life Science Alliance. Congratulations on this interesting work.

Your manuscript will now progress through copyediting and proofing. We acknowledge that you have edited your abstract and, at the proofs stage, we encourage you to improve clarity on the last sentence of your abstract. We suggest: "Identification of a RhoA-ROCK1-independent mechanism by which Rnd3 modulates alpha 5 integrin expression to control cell migration and invasion provides new insights into the molecular basis of pro-malignant properties. These properties that drive the metastatic potential of lung adenocarcinoma may be exploited to target metastatic disease.",

It is journal policy that authors provide original data upon request.

DISTRIBUTION OF MATERIALS:

Again, congratulations on a very nice paper. I hope you found the review process to be constructive and are pleased with how the manuscript was handled editorially. We look forward to future exciting submissions from your lab.

Sincerely,

Sarita Hebbar, PhD
Scientific Editor
Life Science Alliance
<http://www.lsajournal.org>